# The Potential of Naturalistic Eye Movement Tasks in the Diagnosis of Alzheimer’s Disease: A Review

**DOI:** 10.3390/brainsci11111503

**Published:** 2021-11-12

**Authors:** Megan Rose Readman, Megan Polden, Melissa Chloe Gibbs, Lettie Wareing, Trevor J. Crawford

**Affiliations:** Department of Psychology, Lancaster University, Bailrigg, Lancaster LA1 4YF, UK; m.polden@lancaster.ac.uk (M.P.); m.gibbs@lancaster.ac.uk (M.C.G.); l.wareing2@lancaster.ac.uk (L.W.); t.crawford@lancaster.ac.uk (T.J.C.)

**Keywords:** Alzheimer’s disease, mild cognitive impairment, eye tracking, naturalistic eye movement tasks, cognitive impairment

## Abstract

Extensive research has demonstrated that eye-tracking tasks can effectively indicate cognitive impairment. For example, lab-based eye-tracking tasks, such as the antisaccade task, have robustly distinguished between people with Alzheimer’s disease (AD) and healthy older adults. Due to the neurodegeneration associated with AD, people with AD often display extended saccade latencies and increased error rates on eye-tracking tasks. Although the effectiveness of using eye tracking to identify cognitive impairment appears promising, research considering the utility of eye tracking during naturalistic tasks, such as reading, in identifying cognitive impairment is limited. The current review identified 39 articles assessing eye-tracking distinctions between people with AD, mild cognitive impairment (MCI), and healthy controls when completing naturalistic task (reading, real-life simulations, static image search) or a goal-directed task involving naturalistic stimuli. The results revealed that naturalistic tasks show promising biomarkers and distinctions between healthy older adults and AD participants, and therefore show potential to be used for diagnostic and monitoring purposes. However, only twelve articles included MCI participants and assessed the sensitivity of measures to detect cognitive impairment in preclinical stages. In addition, the review revealed inconsistencies within the literature, particularly when assessing reading tasks. We urge researchers to expand on the current literature in this area and strive to assess the robustness and sensitivity of eye-tracking measures in both AD and MCI populations on naturalistic tasks.

## 1. Introduction

In Alzheimer’s disease (AD), the accumulation of intracellular neurofibrillary tangles, extracellular amyloidal protein deposits (senile plaques), and the subsequent disruptions in synaptic transmission (see [1] for a review of AD pathology), results in profound cognitive impairments [2,3]. For example, overall deficits in memory (e.g., recalling recent events), and more specific deficits in language, semantic memory, attention, and visuospatial function characteristically occur in AD [4,5,6,7]. Typically, the diagnosis of AD relies upon a ‘ruling out approach’ in which people undergo extensive physical/neurological assessments, including biochemical analyses (e.g., lumbar punctures), functional brain imaging (e.g., fMRI), and neuropsychological cognitive screening (e.g., the Montreal cognitive assessment (MoCA; [8]), to rule out alternative neuropathologies. Furthermore, diagnosis relies upon subjective reporting of daily capabilities by the person being assessed and, often, a close relative. This approach can be problematic if people do not have a full or accurate understanding of their cognitive capabilities and are unable to accurately articulate these to a health care professional. Consequently, the currently employed diagnostic protocol is not only lengthy and at times subjective, but is also invasive (e.g., lumbar punctures) and costly (e.g., clinical assessment and neuroimaging). These inherent drawbacks have led to a concerted research effort in identifying alternative cost-effective and time-sensitive diagnostic tools.

Eye tracking is a non-invasive advanced technology that provides reliable multifaceted measures of an individual’s saccades (rapid eye movements) whilst performing tasks [9,10,11]. The current evidence suggests that attention is the first non-memory domain to be affected in AD [4]. As attention and oculomotor control are thought to recruit overlapping brain regions [12], saccades are likely to be disturbed by the reductions in inhibitory control and executive function that occur in neurodegenerative disorders [13]. As a result, the utility of low-cost eye-tracking technologies in distinguishing an array of neurodegenerative disorders from their healthy counterparts has received much interest.

The prosaccade task requires participants to perform rapid, reactive saccades towards a suddenly appearing target from a central fixation point ([14], see panel A Figure 1). Interestingly, some evidence has shown that the latency of saccades produced by people with AD are longer than healthy older controls (HOC; [15]). However, alternative research has found no differences in saccadic latency between people with AD and HOC (see [15] for review). Due to these inconsistencies, it appears that prosaccade tasks alone are not sufficiently sensitive to function as an AD diagnostic tool.

Conversely, the antisaccade task has yielded more consistent results. This task requires participants to inhibit a reactive saccade towards a target and instead perform a saccade towards the opposite target-absent location ([16]; see panel B Figure 1). Specifically, whilst antisaccade latencies do not appear to differentiate between people with AD and HOC [15], the frequency of inhibition errors made on the antisaccade task is significantly higher in those with AD [11,17,18]. Moreover, the frequency of inhibition errors on the antisaccade task has been found to be predictive of dementia severity [11,14,17]. Furthermore, whilst HOC correct a large proportion of antisaccade task errors, people with AD often fail to do this, resulting in a higher number of uncorrected errors than HOC [10,11,17,19,20]. The homogeneity demonstrated in the literature suggests that the antisaccade task may be a valid AD diagnostic tool [21].

Although extensive literature has demonstrated the potential, robustness, and sensitivity of antisaccade tasks in distinguishing people with AD from HOC and those with MCI, the task is not without its limitations. The main goal of the antisaccade task is to divert your gaze away from a salient stimulus to a target-absent location. This task requires participants to employ an uncommon eye movement that is often counterintuitive. Therefore, one can argue that the task is low in ecological validity. Recent research has attempted to address these issues by assessing inhibitory control while providing a gaze-directed target and thereby eliminating the antisaccade eye movement [22]. Additionally, research has employed eye-tracking techniques during naturalistic tasks, such as TV watching and reading [23], which typically involve similar inhibitory control capabilities as those employed in the antisaccade task [24]. For example, Forde et al. [25] analysed the eye movements of an individual with action disorganisation syndrome, several people with AD, and HOC whilst they made a cup of tea.

Naturalistic tasks typically involve similar inhibitory control capabilities as those employed in the antisaccade task [24]. For example, when watching a video of two people talking on a busy street, the watcher must remain focussed on the people and avoid distracting background information such as cars. Failure to successfully inhibit background information and focus on the most salient parts could lead to difficulties in deriving meaning and understanding the video. Additionally, free-viewing visual search tasks require participants to freely view a scene without explicit goal-directed instructions. Therefore, this removes the requirement for artificial influences to dictate where participants direct their visual focus (see Figure 1 for exemplar eye movement patterns during naturalistic tasks). Reading tasks also involve inhibitory control processes as the participants are required to direct their gaze to relevant parts of the text while inhibiting excessive regression fixations to previously read text. As these tasks employ similar inhibitory control processes as the antisaccade task, it is possible to assess inhibitory control capabilities while utilising naturalistic and familiar tasks.

In addition, novel lab-based tasks require the participants to quickly adapt, follow instructions, and learn new behaviours to complete these tasks successfully. Critically, there are many factors, such as age, sex, intelligence, and motivation, that may influence an individual’s ability to learn a new behaviour [26]. Intuitively, these factors are likely to influence both neurotypical people and people with neurological impairment, particularly in early stages of the task. Subsequently, altered eye movement behaviours may reflect a lack of task understanding rather than the presence or absence of a cognitive disorder. In contrast, it is likely that naturalistic tasks, such as reading or tea making, will already be familiar tasks to participants, and therefore require little to no explanation of how to complete the task. This removes the increased level of difficulty of having to learn a new task and reduces the likelihood of misunderstanding the task instructions. Subsequently, naturalistic tasks could improve the robustness and ecological validity of eye movement tasks, which in turn will further improve their utility as a diagnostic tool and early indicator of cognitive impairment. Furthermore, naturalistic tasks can result in a more relaxed testing environment that decreases the anxiety that can occur when completing alien tasks. This, consequently, can lead to a more accurate representation of the individual’s cognitive capabilities.

As AD is a progressive disorder, preclinical cognitive decline, known as mild cognitive impairment (MCI), typically precedes AD [27]. MCI occurs when people experience cognitive decline over and above that usually expected with normal aging but below that of AD [28]. The classification of MCI can be further subdivided into amnestic (aMCI) and non-amnestic MCI (naMCI; [29]). Those with aMCI typically display mild memory deficits that do not meet the criteria for dementia, whereas those with naMCI typically have preserved memory but display more general decline (e.g., executive functioning deficits; [30]). The probability that an individual with MCI will later develop AD is much higher than in the general population [31]. More specifically, those with aMCI are at greater risk of developing AD than those with naMCI [32,33,34]. For eye tracking to be an efficacious diagnostic tool, it must also be able to differentiate those with preclinical cognitive decline (MCI) from those with AD and HOC. Concerning this, Wilcockson et al. [35] demonstrated that the antisaccade task can distinguish between MCI subgroups. People with AD and aMCI showed slower latencies and higher error rates than people with naMCI and HOC, and people with aMCI performed more similarly to people with AD than people with naMCI or HOC. This thereby supports the notion that not only are antisaccade tasks sufficiently sensitive to differentiate preclinical cognitive decline from AD [15] but they can also differentiate different manifestations of preclinical decline.

Considering the promise of naturalistic eye movement tasks in the diagnosis of disorders of ageing, a somewhat recent review has concluded that naturalistic eye movement tasks have the potential to successfully differentiate healthy older adults from people with MCI [36]. Specifically, Seligman and Giovanetti [36] highlighted that important eye movement patterns, including fixation location, duration, and saccade magnitude, are highly consistent in HOC, and therefore, are sensitive enough to highlight meaningful alterations indicative of MCI. However, the review focussed primarily on MCI studies and excluded a number of domains, including the literature on reading.

Despite the promise of naturalistic eye movement tasks in distinguishing between people with AD and HOC, research in this area remains limited and underdeveloped. Moreover, although considering the same topic area, Seligman and Giovanetti [36] focussed on the theoretical utility of naturalistic eye movement tasks in people with MCI; therefore, the overlap with the present review is minimal. Subsequently, this review aims to summarise the latest developments in the literature concerning naturalistic eye movement task performance in people with AD, MCI, and HOC. Furthermore, it seeks to establish the utility of naturalistic eye movement paradigms in the diagnosis and assessment of cognitive deficits in both AD and MCI groups. With its potential as an early diagnostic tool, it is hoped this review will spark renewed interest in this field and lead to future developments in this area.

## 2. Materials and Methods

The Non-Interventional, Reproducible, and Open (NIRO) Systematic Review guidelines (V1; [37]) were followed to reduce bias during the development of our search strategy, screening, and the critical appraisal of papers. The NIRO Systematic Review guidelines (V1; [37]) comprise a comprehensive checklist to follow when conducting and writing a review of non-interventional research to ensure transparency and reduce bias.

### 2.1. Data Sources

A comprehensive literature search was conducted on 30th July 2021, using PsycInfo, Academic Search Ultimate, and MEDLINE Complete EBSCOhost databases. These databases, accessed through Lancaster University, were selected to address the multidisciplinary nature of the posed research question. Different search strings for each of the populations (AD, MCI, and healthy older adults) and tasks (naturalistic eye movement tasks) of interest were developed.

The search strings applied for each database differed slightly, due to the inclusion of different dictionary terms that are specific to the databases. Otherwise, the free-text search terms remained consistent and were used to search the titles and abstracts of records in each database. Appropriate free-text search terms were identified during scoping searches. A search string with free-text search terms relating to naturalistic tasks was included to increase the relevancy of records and produce a more manageable quantity of records to screen (see Appendix B for the full search strings used in each database).

During screening, we highlighted that the naturalistic tasks search string applied lacked sensitivity. Specifically, articles involving locomotion tasks, which may be relevant to our research question, were not detected. Thus, we conducted forward and backward citation tracking of key articles [36,38,39,40,41,42,43,44,45,46,47] using Google Scholar. We screened the full text of any articles that appeared relevant based upon their titles and abstracts. The citation tracking was conducted on 5th August 2021. Furthermore, we also highlighted that studies analysing face processing would also be relevant to our research question. Therefore, using the same databases, we formed new search strings, using our previous method, to cover search terms for face processing, as well as locomotion. This second search was conducted on 28 August 2021. A final search was conducted with search terms associated with visual paired comparison and free-viewing tasks. This final search was conducted on 19 October 2021. Forward and backward citation tracking was conducted using key articles identified during the second round of screening [48,49,50,51].

In circumstances in which we could not access the relevant record, Lancaster University Library requested access to these records [48,49,52,53]. Literature that has not been published through traditional means, e.g., conference abstracts known as grey literature, is often excluded from large databases [54]. Specific grey literature searches are often conducted when collating the literature for a systematic review [55]. To the best of our knowledge, this review is the first of its kind to specifically focus on naturalistic eye movement tasks, therefore, we did not conduct a grey literature search. However, we recognise that performing grey literature searches are important to prevent publication bias and so we encourage future reviews that aim to build upon this research review to include grey literature.

### 2.2. Screening

The .ris files downloaded during the final search of each database were exported into the reference managing software Zotero. The records were de-duplicated by hand using a Microsoft Excel spreadsheet, which was also used during screening. Of the 1011 records identified from our chosen databases, 270 (26.71%) of these were duplicates and were removed prior to screening.

#### Inclusion Criteria

In line with the preferred reporting items for systematic reviews and meta-analyses (PRISMA) systematic review guidelines [56], the inclusion criteria applied were: (1) full length, English language original studies (e.g., not reviews or book chapters); (2) peer-reviewed articles; (3) the study included an AD/MCI group without comorbidities or other neurological disorders and a relevant control condition; (4) use of naturalistic eye movement tasks; (5) reported statistics for the comparison of eye movements between AD/MCI and HOC.

Previous research has shown that many individuals with AD present with comorbidities. For example, many individuals with AD have anxiety disorders [57], and around 30% of individuals with AD present with comorbid depression [58]. Moreover, some individuals with alternative neurological conditions, including Parkinson’s disease (PD; [59]) and multiple sclerosis [60], present with comorbid cognitive impairment and, in some cases, comorbid dementia. Interestingly, additional research has shown that these comorbidities and alternative neurological disorders can independently substantially influence naturalistic eye movement behaviours. For example, a somewhat recent meta-analysis concluded that individuals with depression show reduced maintenance of gaze towards positive stimuli, and anxious individuals showed difficulty disengaging from threatening stimuli during visual search tasks [61]. Moreover, Stock et al. [62] observed that individuals with PD fixate on words for a greater duration and make a greater number of regressions when reading. As these morbidities and alternative neurological disorders can independently influence naturalistic eye movement behaviours, if one were to analyse naturalistic eye movement behaviours in individuals with AD with these comorbidities, it would be difficult to parse apart the influence of AD from the influence of the comorbidity. As this review sought to analyse the potential utility of naturalistic eye movement tasks in the specific diagnosis of AD/MCI, it is important to strive to reduce the likelihood of including participants with these comorbidities.

Here, naturalistic eye movement tasks were defined as those tasks that either (a) incorporate goal-directed paradigms with naturalistic stimuli (e.g., a prosaccade task in which participants are instructed to perform a saccade towards an object within a naturalistic scenes), (b) tasks in which stimuli was presented for a minimal duration, 5 s, that enabled participants to engage in free unrestricted visual exploration (e.g., unrestricted static image search and visual paired comparison tasks), or (c) tasks that are the same as (e.g., making a cup of tea or navigating an environment) or closely mirrored (e.g., virtual reality) tasks undertaken in a normal daily life setting.

In this review, we deemed prosaccade tasks as a naturalistic paradigm as they replicate eye movements frequently performed in daily life. For example, if individuals are asked to “look at this” or “look over here”, they subsequently perform a prompted goal-directed saccade similar to that employed in prosaccade tasks. In contrast, antisaccade tasks were excluded from this review due to antisaccade eye movements being artificial by nature and unintuitive.

Each paper’s titles, abstracts, and full texts were screened simultaneously by the same two reviewers. The reviewers completed screening separately, so were blind to the other’s decisions until all of the records had been screened. The level of agreement between the two reviewers was 97.36%. When inconsistencies in rating arose, a third reviewer was involved in making the final decision (see Appendix A for decision log; see Figure 2 for a pictorial depiction of the search and screening process). Of the 832 records screened, 793 (95.31%) were excluded due to failure to meet our inclusion criteria. All of the articles that passed the full-text screening phase were checked for retraction using the Retraction Watch Database (25 November 2021, http://retractiondatabase.org/). Of the 39 papers that passed through full-text screening, 27 studies had a singular AD patient group, 7 studies had a singular MCI group, and 5 study had both an AD group and an MCI group.

Resultantly, 39 papers passed through full-text screening and were identified as being relevant to our research question [25,38,39,40,41,42,43,44,45,46,47,48,49,50,51,52,63,64,65,66,67,68,69,70,71,72,73,74,75,76,77,78,79,80,81,82,83,84,85].

### 2.3. Data Extraction

The following data was extracted from the papers that passed the screening: the number of participants; participant groups; mean age of each participant group; study design; criteria used for AD or MCI classification; cognitive assessments used; eye movement task; type of stimuli used; direction of the main effects; any reported means and standard deviations (SDs); effect sizes; and any relevant conclusions made by the authors (see Appendix C for the full study details).

### 2.4. Quality Assessment

To determine the credence that should be given to individual studies we assessed the quality of the papers that were analysed. The Downs and Black checklist [86] is designed to allow the quality assessment of interventional research, including assessments of internal and external validity, reporting, and power. Currently, no specific tools for assessing the risk of bias in non-interventional research have been developed. Therefore, we modified Downs and Black’s [86] risk of bias tool to suit our purposes (see Appendix D). To ensure the checklist was valid for the intended use, items 4, 7, 12, 13, 14, 15, 16, 17, 19, 21, 22, 23, and 24 were removed. Furthermore, items 9 and 26 were edited to apply to the excluded participants (rather than patients’ attrition) and item 27 was edited to address whether any justifications were made for the sample size (rather than only a power analysis). We also modified the ratings responses from yes and no (scoring one or zero points, respectively) to yes, partial, and no (scoring two, one, and zero points, respectively).

Quality assessments were carried out on all of the papers that passed through full-text screening. The level of agreement between the two raters was 78.30%. In circumstances in which there was a disagreement between the raters, a third rater made the final decision. The maximum score for our modified checklist was 26 (*M* = 17.31; range 7–23; see Table 1 for the risk of bias scores for each paper; see Appendix A for the full quality assessment ratings of each paper, including the scores for each item).

## 3. Results

This literature review revealed that studies examining naturalistic eye movements in people with AD, MCI, and healthy older controls can be broadly classified into four domains; reading tasks, goal-directed paradigms with naturalistic stimuli (e.g., goal-directed saccades towards naturalistic stimuli), paradigms that are naturalistic by nature (e.g., free image viewing or visual paired comparisons), and paradigms including or simulating everyday activities (e.g., making a cup of tea or navigating an environment). Importantly, the eye movement behaviours facilitated by these four domains of literature are somewhat distinct. That is, during reading tasks, the participants typically perform highly specialised eye movement patterns, including saccades, fixations, and regressions, mediated by the text they are reading [87]. These highly specialised eye movement patterns are largely distinct from the free exploratory saccades and fixations typically performed during free visual search tasks. Due to the distinct nature of these domains of research, drawing parallels between the obtained results is somewhat difficult and arguably invalid. Subsequently, the results of such studies will be presented separately.

### 3.1. Reading Tasks

Of the 39 studies that met the inclusion criteria, 9 analysed eye movement behaviours during the completion of reading tasks [39,40,41,42,43,44,47,52,75].

All except one of the reading task studies compared eye movement behaviours in those with AD to HOC of comparable age. The remaining study compared eye movement behaviours of those with MCI to HOC. Typically, these studies tracked eye movement behaviours whilst the participants read, either silently or aloud, short sentences, passages, or single words. In doing so, both Fernández et al. [39] and Fernández et al. [40] observed that when reading single-sentence texts, people with AD overall made significantly more fixations than HOC. Similarly, Lueck et al. [52] observed that people with AD made significantly more saccades. Alternatively, Yong et al. [47] observed that when reading text passages comprising three sentences, the overall number of fixations made by people with AD did not significantly differ from HOC. Furthermore, Lueck et al. [52] observed that, in a given time frame, people with AD read a significantly smaller portion of text compared with HOC; thereby indicating that people with AD have a slower reading speed. Moreover, the proportion of text read was significantly correlated with the degree of dementia severity amongst people with AD.

However, some inconsistencies were noted. For example, both Fernández et al. [39] and Lueck et al. [52] observed that people with AD make an increased number of first-pass fixations (i.e., the initial reading consisting of all forward fixations on a word) than HOC. Supporting this observation, Fernández et al. [75] also observed that people with AD made more first-pass fixations compared to the controls. In contrast, Fernández et al. [40] observed that people with AD make fewer first-pass fixations than HOC. Similarly, whilst Fernández et al. [39] observed that people with AD skip significantly more upcoming words than HOC, Fernández et al. [40] observed that people with AD skipped fewer upcoming words than HOC. However, more consistently, Fernández et al. [39], Fernández et al. [40], Fernández et al. [75], and Lueck et al. [52] observed that people with AD make significantly more second-pass (i.e., re-reading a word) and regression (i.e., regressing to a previously read word) fixations than HOC.

Globally, all of the studies measuring fixation duration, with the exception of one, observed that fixation duration was substantially longer in people with AD compared to HOC [40,41,42,43,48]. Moreover, in HOC, as the predictability of upcoming words increased, fixation duration decreased, whereas this effect was not observed in people with AD [40,41,42]. The exception, Lueck et al. [52], observed that the average saccade duration did not differ between people with AD and HOC. When reading highly predictable sentences, there is typically a word at which not only the next word, but the entire sentence becomes available to the visual system. On reading this word, fixation duration significantly decreases in HOC. However, in people with AD, fixation duration increased on reading this word [43]. Finally, word frequency influenced fixation duration, in that fixation duration was significantly increased for longer words in both HOC and people with AD [40,41].

Some studies analysing eye movement behaviours during prosaccade tasks have observed that people with AD typically produce hypometric saccades (saccades of reduced amplitude) compared to HOC (e.g., [88]). Fernández et al. [39], Fernández et al. [40], and Fernández et al. [75] replicated this finding during reading tasks, observing that the mean outgoing saccade amplitude of people with AD was significantly smaller than HOC.

Concerning people with MCI, Fraser et al. [44] observed that people with MCI made significantly fewer first-pass fixations and significantly more later-pass fixations than HOC. Thus, people with MCI tend to skip a greater proportion of words and make a larger number of fixations and saccades back to these words.

### 3.2. Studies Employing Goal-Directed Paradigms with Naturalistic Stimuli

Of the included studies, seven studies incorporated goal-directed paradigms with naturalistic stimuli [66,67,68,71,74,76]. Moreover, whilst Coco et al. [69] and Shakespeare et al. [73] incorporated tasks that were naturalistic by nature (e.g., free image search), the duration of the image presentation was insufficient for the participant to engage in free visual exploration. Thus, these tasks could not be classified as naturalistic in accordance with the definitions prescribed here. Therefore, the results obtained by Coco et al. and Shakespeare et al. [69,73] will be presented here.

Vallejo et al. [66] analysed eye movements during a Go-NoGo visual search task of naturalistic scenes. People with AD were slower at detecting targets than HOC but the mean saccade fixation time of people with AD did not differ from HOC. Interestingly, whilst Vallejo et al. [66] observed that people with AD were slower at detecting targets than HOC, both Lenoble [76] and Bourgin et al. [74] found that the latency of saccades made to naturalistic stimuli by people with AD were comparable to healthy younger [76] and older controls [74,76]. Moreover, Vallejo et al. [66] also found that people with AD demonstrated an impaired ability to detect targets in central positions compared with HOC, but demonstrated a preserved ability to detect targets in the peripheral positions. People with AD made more fixations to the periphery of the display and less to the centre of the display when compared to HOC, which is indicative of an inability to voluntarily direct attention to central cues.

Further buttressing these observations, Boucart et al. [67] and Boucart et al. [68] observed that when asked to perform a saccade to the naturalistic scene containing an animal, as opposed to a competing scene, people with AD were less accurate than both younger and older adults. More specifically, Boucart [68] observed that the first saccades of younger controls were more likely to land in the region of interest (ROI) than those of HOC or people with AD. An analysis of the characteristics of saccades made towards naturalistic scenes containing specific objects revealed that the latencies [68,74], amplitude [68], and duration of saccades [68] produced by people with AD are comparable to controls.

In contrast, Lenoble et al. [71] and Lenoble [76] observed that people with AD made a comparable number of errors to HOC when asked to make saccades towards objects in naturalistic scenes. However, interestingly, whilst Lenoble et al. [71] observed that people with AD made significantly more errors than younger controls, Lenoble et al. [76] observed that younger controls made a comparable number of errors to people with AD. The accuracy of the first saccade produced by HOC was not influenced by whether the target was presented on a congruent or incongruent background. In comparison, people with AD were more accurate at detecting targets when they were presented on an incongruent background. When the participants were asked to perform a saccade to the congruent image (the image in which the target is presented on a congruent background), the younger controls reached the target on the first saccade more often than HOC, and HOC reached the target on the first saccade more often than people with AD.

In a free-viewing task that allowed insufficient time for the participant to engage in free visual search, Coco et al. [69] showed the participants a stream of naturalistic images belonging to different semantic categories (e.g., bathroom, beach, or kitchen). The participants were asked to state which of two scenes (one novel and one original from the same semantic category) they had seen before. The frequency of images falling into a semantic category was systematically varied to induce semantic interference. During free image viewing, the fixation patterns of HOC were significantly less focussed than people with MCI (higher fixation entropy), thereby suggesting that HOC had a wider spread of attention than people with MCI. During the test phase (identifying which of the two presented scenes was novel), the fixation patterns of both HOC and people with MCI were significantly less focussed for correctly identified scenes, as opposed to incorrectly identified scenes (increased fixation entropy). Moreover, the fixation patterns became less focussed as semantic interference increased in both people with MCI and HOC; however, this effect was significantly reduced in people with MCI. Interestingly, scan pattern similarity (between free-viewing and recognition phases) was higher when the scene was recognised in both HOC and people with MCI. The reliance on low-level visual features of scenes displayed by people with MCI was comparable to that of HOC. These results therefore suggest that semantic interference effects are present in MCI populations, but at a lower potency than within healthy adults. However, the authors noted that the observed semantic interference effects may have been skewed by individual differences within the people with MCI [69].

Moreover, in another free-viewing task of insufficient time, Shakespeare et al. [73] also observed that people with AD made fewer fixations within the ROI than HOC when scanning scenes for specific objects. Furthermore, people with AD took more time to make their first fixation within the ROI than HOC. Despite these differences between people with AD and HOC, saccade amplitude and the average distance of fixation away from the centre of the image did not differ between people with AD and HOC. Furthermore, considering the application of task strategies, Shakespeare et al. [73] observed that HOC adapted their scan path based on the study task to a greater extent than people with AD. A reduction in this ability may in part be reflective of executive functioning deficits observed in AD [13] and may demonstrate a reduced ability to employ task-appropriate scan patterns and alter task strategy as quickly as HOC.

### 3.3. Studies Employing Naturalistic Tasks

Interestingly, this literature search revealed that eye movement tasks that are naturalistic by nature somewhat vary. Subsequently, to facilitate direct comparisons between similar studies, we have further subgrouped naturalistic tasks into the following sections: Eye Movement Behaviours during Static Image Search, Eye Movement Behaviours during Visual Paired Comparison Tasks, Eye Movement Behaviour during Every-day tasks and Real-life Simulations, and Eye Movement Behaviours during Facial Processing.

#### 3.3.1. Eye Movement Behaviours during Static Image Search

Seven of the included studies analysed eye movement behaviours during static image search tasks [38,45,63,65,70,71,72]. Interestingly, based on our search, static image search tasks appear to be the most common naturalistic eye movement tasks employed when analysing eye movement behaviours in AD/MCI.

Brandão et al. [63] analysed eye movements during free recall of an important life event whilst relevant or irrelevant visual cues (images and sentences) were presented. In doing so, Brandão et al. [63] observed that HOC fixated on relevant images longer than irrelevant images, however, this effect was not observed for people with AD. In general, people with AD fixated their gaze on the screen (as opposed to looking at the experimenters’ face) more when visual cues were present, irrespective of their relevance, hereas HOC attended to the screen more only when the visual cues were relevant. Comparably, considering both people with MCI and AD performance during visual search of a naturalistic scene, Dragan et al. [38] observed that the eye movement search patterns of people with AD were significantly less focussed than those of HOC. The eye movement search patterns of people with MCI were also less focussed than HOC, but this difference was not significant. Furthermore, people with AD made significantly more fixations before finding the target object than both people with MCI and HOC.

Mosimann et al. [65] observed that when visually exploring a clock face, the time to first fixation within the ROI was significantly longer in people with AD. Furthermore, they found that fixation durations of people with AD were longer than HOC. In addition, Mosimann et al. [65] also observed that people with AD saccades were significantly shorter than HOC.

Moreover, Daffner et al. [70] found that when viewing photographs containing an incongruous element (for example, a lion in a classroom of children), people with AD looked at significantly fewer ROIs for a significantly shorter duration than HOC. However, this pattern of altered eye movement behaviours was not observed in all incongruent images, thus, this effect may be contingent upon the stimuli presented. Comparably, Oyama et al. [72] observed that people with dementia fixated on ROIs for a shorter duration than people with MCI and HOC. Furthermore, fixation duration correlated with scores on the Mini-Mental State Examination (MMSE). Specifically, people who scored higher on the MMSE also presented longer fixation durations. Similarly, Lenoble et al. [71] observed that when presented with a naturalistic image containing a congruent or incongruent object, HOC looked at pictures containing an incongruent object significantly longer than people with AD.

This enhanced distractibility displayed by people with AD and MCI [38,63,65,70,71,72] is likely linked to well-known inhibitory control deficits. Inhibitory control deficits in AD and MCI populations are evident on established eye movement paradigms, such as the antisaccade task [16], and result in disrupted eye movements and a reduced ability to inhibit distracting stimuli. Therefore, the above studies support previous findings surrounding inhibitory control deficits in people with AD and MCI.

When considering the characteristics of the saccades performed during a naturalistic static image search, LaBar et al. [45] failed to observe any differences in saccade latency between individuals with AD and HOC. In this task, the participants were presented with pairs of visual scenes that ranged from emotionally negative to neutral and instructed to view them however they wished.

#### 3.3.2. Eye Movement Behaviours during Visual Paired Comparison Tasks

The visual paired comparisons (VPC) task has a proven sensitivity to memory decline [89]. Typically, during the VPC task, participants are first presented with a visual stimulus for a fixed period of time (familiarisation phase). Following a delay, participants are presented with a pair of stimuli, one that is the same as the familiarisation stimulus and one that is new (test phase; [90]). As the participants are not instructed where to direct their gaze during both the familiarisation and test phases, participants will engage in free visual search. Consequently, even if the visual stimuli presented are artificial (e.g., line drawings), the visual search strategy engaged by the participant is naturalistic by nature.

Six of the included studies employed VPC comparison tasks. More specifically, of these studies two included naturalistic visual stimuli, two included artificial stimuli, and two analysed VPC performance longitudinally using artificial stimuli. Both Chau et al. [77] and Lagun et al. [78] assessed performance on the VPC task, incorporating artificial stimuli. Chau et al. [77] first presented participants with a slide containing four novel images. This was followed by two further slides containing two novel images and two repeated images. The relative fixation time was calculated by dividing the fixation time to the novel images by the total fixation time for all four of the slide images. In doing so, Chau et al. [77] found that people with AD showed lower relative fixation times when viewing novel images than on repeated images compared to HOC. In addition, a reduced relative fixation time was associated with lower MMSE task scores. Interestingly, Lagun et al. [78] also assessed VPC task performance in people with MCI. From this, Lagun et al. [78] found that VPC performance can effectively distinguish between people with AD, MCI, and HOC. Specifically, machine learning demonstrated an accuracy of 87%, sensitivity of 97%, and specificity of 77% when distinguishing participant groups.

In contrast, both Crutcher et al. [79] and Haque et al. [80] incorporated images of naturalistic scenes during the VPC. Assessing people with MCI and HOC, Crutcher et al. [79] varied the delay interval (2 s or 2 min) between the initial viewing of the image and the test trial (in which the repeated image and novel image were presented simultaneously). Interestingly, at the 2 s delay, the participants’ viewing behaviour was comparable; a novel image preference of 71% was observed across the groups. However, when the delay between images increased to 2 min, the viewing preference for the novel image was significantly reduced only in people with MCI. This finding demonstrates that a delay period during the VPC highlights viewing pattern distinctions in cognitively impaired populations compared to healthy adults.

Haque et al. [80] looked at VPC in people with AD, MCI, and HOC. The participants were asked to view coloured images of naturalistic scenes with no explicit instructions. After an initial viewing, the participants were presented with the image once again but with either an item removed from the scene or an item added to the scene. The ROI were defined as the location from where the item was removed or added. For people with AD and MCI, the time spent viewing the ROI and number of fixations to the ROI was significantly lower compared to HOC. Thus, there were clear performance differences between cognitively impaired individuals and HOC. The results from the above studies indicate that visual scanning behaviour, specifically novelty preference, varies between HOC and people with AD and MCI, highlighting key and robust markers for cognitive impairment.

Two studies utilising a VPC methodology analysed performance longitudinally [81,82]. In doing so, both Zola et al. [81] and Nie et al. [82] corroborate with the aforementioned findings that fixation duration on novel stimuli was significantly shorter in MCI and AD than HOC. Echoing the findings of Crutcher et al. [79], Nie et al. [82] found that novelty preference only differed significantly between people with MCI and HOC when the delay period was 2 min, but not 2 s. Moreover, Nie et al. [82] found that this difference remained significant at a two-week follow-up.

#### 3.3.3. Eye Movement Behaviours during Facial Processing

Three of the included studies analysed eye movement behaviour in people with AD, MCI, and HOC while processing facial stimuli.

Kawagoe et al. [50] had people with MCI and HOC judge whether two images (faces or houses) were the same or different, and indicate which of the two images, if any, had previously been presented. When judging whether the images were the same or different, HOC fixated on the eye and nose longer than any other facial landmark, however, this effect was not observed in people with aMCI. In contrast, Kawagoe et al. [50] found that when judging whether an image had been previously presented, the observed fixation pattern did not differ between HOC and people with aMCI.

Concerning visual exploration of face stimuli, Ogrocki et al. [48] observed that, in general, people with AD fixated less on the presented faces, particularly the eye regions. People with AD also spent less time exploring different facial regions, and rather spent more time focussing on specific areas of the face than HOC. Similarly, during a passive face-viewing task, McCade et al. [49] observed that people with aMCI, naMCI, and HOC all fixated on the eye region significantly longer than other facial regions.

The inconsistencies in observed fixation duration patterns across the studies may in part be a consequence of the differential task demands. This assumption is further supported by McCade et al. [49] who observed that fixation duration patterns differed as a consequence of the emotionality of the face stimuli. Specifically, for disgusted and angry faces, participants fixated on the eye region less when compared to neutral faces. Moreover, participants fixated more on the mouth region of disgusted and surprised faces compared to neutral faces.

### 3.4. Eye Movement Behaviours during Every-Day Tasks and Real-Life Simulations

Five [25,46,51,64,83] of the studies included here analysed eye movements during real-life simulations. Specifically, two of such studies analysed eye movement behaviours during every-day tasks [25,51], and the remaining two employed tasks that simulated real-life situations. [44,64].

Forde et al. [25] tracked the eye movements of a person with action disorganisation syndrome, a person with AD, one HOC, and one younger control whilst they made a cup of tea. Interestingly, Forde et al. [25] observed that the person with AD made a comparable number of fixations of equivalent fixation duration to younger and older controls. More specifically, the proportion of task-relevant and task-irrelevant fixations did not differ between the HOC, young control, and people with AD. In HOC, young controls, and people with AD, 10–15% of fixations were to relevant objects that were to be used in the next stage of the tea-making tasks.

Similarly, Yong et al. [51] analysed participants’ eye movements as they walked to a visible destination that was either cued with a contrast cue (a black box above the target door handle), both a contrast cue and a motion cue (the black box and a rotating light), or no cue. From this, Yong et al. [51] failed to observe any differences in target fixation or fixation duration between people with AD, posterior cortical atrophy (PCA), and HOC. The only circumstance in which eye movement behaviours of people with AD differed from HOC was in the contrast cue paired with motion cue condition. Under this condition, people with AD made significantly longer fixations on the target location compared with the no cue condition. More advanced AD was associated with orientation to lower visual space. Similarly, Suzuki et al. [83] found that the durations of fixations across all of the locomotion tasks (e.g., walking through corridors, walking up or down stairs, walking through a room with or without an obstacle) did not significantly differ between the AD patient and HOC.

Mapstone et al. [46] employed a driving simulation task where participants passively viewed three street driving simulations from the driver’s perspective. From this, Mapstone et al. [46] found that the total amount of fixations, duration of fixations, and percentage of fixations within the region of interest (ROI; a focus on the street in the direction of travel) did not differ between people with AD and HOC. By contrast, younger controls made more total fixations and fixations to the ROI compared to older adults. This suggests that older adults and people with AD are unable to covertly attend to distractors in their peripheral vision and, instead, direct their full visual attention using overt eye movement to peripheral distracters when driving.

Whilst Mapstone et al. [46] focussed solely on people with AD and HOC, Davis and Sikoriskii [64] also included a preclinical decline (MCI) group in their study. However, during the analysis, people with early-stage AD and MCI were merged into one AD experimental group, thus, we cannot fully ascertain the dissociation between eye movement in those with AD and MCI on these tasks. Davis and Sikoriskii [64] had participants actively navigate their way through a simulated senior retirement community. Here, they identified visual cues embedded in the virtual environment as the ROIs. These cues were classed as ‘salient’ if they acted as landmarks towards the desired location and non-salient if they were irrelevant. Employing this methodology, Davis and Sikoriskii [64] observed that people with AD/MCI made significantly fewer fixations that were also shorter in duration to salient cues compared to HOC. Comparatively, for non-salient cues, people with AD made significantly more fixations than HOC. However, the durations of fixations, for non-salient cues, did not differ between AD and HOC. These eye movement patterns suggest that people with AD/MCI showed a reduced ability to distinguish salient from non-salient cues when navigating an environment and struggle to inhibit task-irrelevant stimuli.

### 3.5. Analyses of the Specificity and Sensitivity of Eye Movements in Diagnostic Practices

Previous research has demonstrated the potential of machine learning to aid in the screening and early diagnosis of neurodegenerative disorders [91] Subsequently, machine learning models built on naturalistic eye-tracking data from people with AD and MCI could offer a non-invasive screening tool to aid with the early detection of cognitive impairment. In this current work, we identified five papers [78,80,81,84,85] that utilised machine learning techniques and conducted an area under the curve (AUC) analysis to decipher the specificity and sensitivity of naturalistic eye movement tasks in differentiating people with AD, MCI, and HOC.

Considering the utility of reading tasks, Fraser et al. [84] tracked participants’ eye movements whilst reading, either silently or aloud, before they completed a comprehension task. Fraser et al. [84] found that the best classification result, achieved by combining eye tracking, speech, and comprehension questions measures) (AUC = 0.88, accuracy = 0.83) outperforms a classifier trained on neuropsychological tests (AUC = 0.75, accuracy−0.65). Thus, indicating that eye tracking and audio recording during reading tasks could aid in the classification of cognitive impairment and may prove more successful than current neuropsychological tests.

Alternatively, considering static image search tasks, Barral et al. [85] asked people with AD and HOC to perform the Cookie Theft picture description task. This required participants to scan a line drawing and verbally describe the scene while their eye movements and speech were recorded. Interestingly, here, Barral et al. [85] observed that eye-tracking data combined with machine learning models can successfully distinguish people with AD and HOC (AUC = 0.73). This model was further improved by combining the eye tracking and speech data (AUC = 0.80).

Lagun et al. [78] assessed people with AD, MCI, and HOC on a VPC task using abstract images. From this, Lagun et al. [78] found that when fixations, saccades, and re-fixations during the VPC task are modelled in tandem with the support vector machines (SVMs) algorithm, people with MCI can be distinguished from HOC with accuracy of 87%, sensitivity of 97%, and specificity of 77%. Consequently, this study provides strong support that eye movement patterns during VPC tasks can distinguish people with MCI and HOC and that machine learning could aid in the automatic detection of cognitive impairment.

Utilising a longitudinal methodology, Zola et al. [81] analysed whether VPC is reflective of cognitive decline. Specifically, an AUC analysis showed that all but one participant who had a novelty preference of less than 50% on the task at initial testing changed in their diagnosis within the 3-year interval of testing. Participants who scored between 50% and 67% were at less risk. Critically, those who scored more than 67% were at a zero risk of further cognitive decline regardless of whether they were initially categorised as HOC or aMCI. Therefore, the VPC task had the capability to predict the participants who would change in their diagnosis (regardless of whether they were HOC or aMCI) before the diagnosis was changed clinically. Critically, when assessing the novelty preference after either a 2-s or 2-min delay, Nie et al.’s [82] AUC analysis showed that novelty preference scores of 0.605 in the 2-min delay task could effectively distinguish MCI and HOC (70% accuracy, 72% specificity, and 53% sensitivity). In a 12-month follow-up, nine participants had progressed to MCI. Those participants whose novelty preference score fell below the 0.605 cut-off point at the initial testing showed significantly greater cognitive decline at the 12-month follow-up.

Similar to the VPC task, Haque et al. [80] assessed people with AD, MCI, and HOC on a visual-spatial memory task in which a familiarised presented image was altered by the removal or addition of an item. Using MoCA scores as a comparison, Haque et al. [80] found that performance on the visual-spatial memory task achieved an AUC of 0.85 (sensitivity = 0.83, specificity = 0.74). Moreover, when compared with disease status, the model achieved an AUC of 0.85, sensitivity of 0.85, and specificity of 0.75. Overall, the above studies appear to provide support that performance on naturalistic eye-tracking taskseye-tracking tasks can aid in the classification and identification of AD and MCI status with high sensitivity.

## 4. Discussion

Naturalistic eye-tracking tasks present a means of examining subtle changes in daily functioning [92] which, critically, can be indicative of an individual at the early stages of or at risk of developing AD [93]. Moreover, they allow an individual’s cognitive function to be assessed when performing natural tasks which inherently require more complex cognitive interactions than traditional eye-tracking paradigms [36]. Furthermore, by their naturalistic nature, they are more familiar and consequently less stressful for the participant. Therefore, there is great potential for naturalistic tasks as an early diagnostic tool.

This review sought to summarise the latest developments in the literature concerning naturalistic eye movement task performance in people with AD, MCI, and HOC. In doing so, it sought to establish the utility of naturalistic eye movement paradigms in the diagnosis and assessment of cognitive deficits. Interestingly, this review highlighted that naturalistic eye movement behaviours in people with AD and MCI have gained consistent research interest from the early 2000s until the present day. Thus, highlighting the theoretical and practical relevance of the analysis of naturalistic eye movement behaviours in people with AD and MCI.

A quality assessment of all of the included papers revealed that the majority of papers suffered from a moderate risk of bias (score rating: 15–20; [38,39,40,41,42,43,44,46,47,48,49,51,52,63,64,65,66,68,69,70,71,72,73,74,75,76,77,78,79,81,82,84,85]), with two receiving a particularly high risk of bias score (score rating: 0–14; [25,45,80,83]). Here, bias refers to factors that can systematically affect the observations and conclusions of a study [94]. Subsequently, some examples of sources of bias include problems with the comparability of the criteria used to select samples (selection bias; [95]), problems with the measurement of outcomes (detection bias; [94]), and problems with whether research is published or not (publication bias; [96]). Interestingly, the three most common sources of bias found within the included papers were: failure to describe the characteristics of participants lost to exclusion, failure to take into account participants lost to exclusion in analyses, and no justification for sample sizes (see Appendix A for quality assessment ratings of each paper). Therefore, as many of the papers included here suffer from a moderate to high risk of bias, a certain level of caution should be assumed when considering the potential application of the findings highlighted here.

Some promising patterns are visible when considering the results across the methodologies. For example, the lack of a word predictability effect in people with AD during reading [40,41,42,43], a lack of scanpath modulation during visual search [73], difficulty with repeated-trial target detection [38], as well as reduced novelty preference in the visual paired comparison task [77,78,79,80,81,82] are indicative of impaired memory recognition. Whereas patterns of increased second-pass fixations and regressions [39,40,52] during reading, an inability to inhibit task-irrelevant stimuli during navigation [46,64], cued conversation [63], saccadic choice tasks [68], and clock reading [65], alongside a decreased ability to detect targets during visual search [38,63,65,66,67,68,70,71,72,73] and real-life simulations [46,64] suggest the presence of an impairment in selective attention (attending to relevant stimuli and inhibiting irrelevant stimuli) amongst AD populations. These observations are consistent with prior literature that suggests visual memory recognition is impaired even at early stages of AD [97], as well as reviews that argue impairments in selective and divided attention, but not sustained attention, are present in the early stages of AD [4]. These results are also consistent with deficits in inhibition that are present when people with AD perform the antisaccade task [35].

The consistency between the observations obtained from naturalistic eye-tracking paradigms with previous literature ultimately suggest that naturalistic tasks may have the capacity to reliably distinguish between AD and HOC on the basis of recognition and attention deficits. Corroborating this, AUC analysis revealed that reading tasks [84], VPC, and similar visuospatial tasks [78,80,81], and static image search tasks [85] have a good to excellent diagnostic accuracy [98] when applied with machine learning to differentiate people with AD, MCI, and HOC.

It should be highlighted that although all of the studies included in this review employed naturalistic tasks or goal-directed paradigms with naturalistic stimuli, the tasks themselves remained lab-based and arguably somewhat contrived. Whilst it is possible that AD and MCI diagnostic tests may occur in naturalistic environment (e.g., an individual’s home or care home), diagnostic tests for AD and MCI are most likely to occur in a clinical hospital setting. Therefore, emphasis should be placed on employing familiar, daily living tasks during diagnosis. This being said, a distinction should be drawn in the literature between tasks that employ naturalistic tasks, or goal-directed paradigms, with naturalistic stimuli in a lab-based setting and tasks which are naturalistic and occur in naturalistic settings (making a cup of tea). When drawing this distinction, it is clear that further research analysing people with AD and MCI’s eye movements during naturalistic tasks is required. Specifically, only 2 [25,51] out of the 39 studies used naturalistic tasks outside of a lab-based environment; thus, demonstrating the lack of current literature assessing AD and MCI eye movements during natural daily activities, such as tea-making. Furthermore, five of the eight studies analysing eye movement behaviours during reading were conducted by the same research group and employed largely the same methodology. Resultantly, there is a lack of diversity in the literature that subsequently limits the reliability and validity of any conclusions that can be drawn.

Although some promising patterns in eye movement behaviours have been highlighted, the presence of inconsistencies in observed eye movements both within and across methodologies raises a concern as to the sensitivity of naturalistic eye-tracking methodologies as a diagnostic tool. Specifically, concerning reading paradigms, the number of overall fixations, first-pass fixations, and skipping frequency made by people with AD compared to that of HOC is inconsistent amongst the included studies. For example, whilst some observed an increase in the number of first-pass fixations in people with AD [39,52], others reported fewer first-pass fixations [40] compared with HOC. Similarly, in employing real-life simulation methodologies, some (e.g., [64]) have observed that increases in fixation frequency and duration occur in AD, whereas others [46] have failed to observe a difference between those with AD and HOC. Comparably, whilst some studies employing static image search methodologies observed alterations in eye movement characteristics in AD [68,73], others (and on occasion the same paper) failed to observe the alterations in eye movement characteristic between AD and HOC [45,68,73,74]. These inconsistencies may imply that naturalistic eye movement tasks are insufficiently sensitive to serve as an effective diagnostic tool.

However, some of these inconsistencies may be explained by methodological variations, for example, the active opposed to passive nature of the tasks applied in Davis and Sikoriskii and Mapstone et al. [46,64]. Similarly, concerning static image search methodologies, the only study that required the participant to passively view stimuli with no additional goal-directed task was also the only study to observe no significant differences in the eye movement behaviour between people with AD and HOC [45]. This further highlights and supports the reasoning that the differences in the results may be due to varying methodologies. Therefore, eye movement variations may be an artefact of the task employed as opposed to the insensitivity of naturalistic eye movement tasks. The current literature review included multiple studies with varying methodologies that differ in their complexity and task difficulty. Goal-directed, unfamiliar tasks are likely to prove more taxing than free-viewing tasks, particularly for individuals with cognitive impairment. Further, unfamiliar, lab-based assessments require the participant to first understand the task instructions and then quickly learn how to perform the task successfully, increasing the difficulty and complexity of the task. Familiar everyday tasks, such as reading, tea making, and free viewing of scenes, do not require this learning process and allow for a more natural assessment of participants’ eye movements. However, results from reading and tea-making tasks may not be sufficiently sensitive to distinguish between people with AD and HOC [25]. The increased level of complexity of goal-directed eye movement tasks may be required to robustly identify cognitive impairment in preclinical stages.

Further, this review highlighted just three studies which utilised facial stimuli, and among these studies there is a high degree of variability in their findings. Whilst Ogrockie et al. (recruiting people with AD; [48]) and Kawagoe et al. (recruiting people with aMCI, [50]) reported deficits in face scanning amongst these groups, McCade et al. [49] observed comparable face scanning in aMCI, naMCI, and HOC groups. Therefore, the limited and highly variable data makes forming conclusions regarding the efficacy of face processing paradigms as an early diagnostic tool limited. Moreover, given that both Ogrockie et al. [48] and Kawagoe et al. [50] reported similar facial processing deficits in AD and aMCI populations, it is unclear whether these tasks have the sensitivity to be able to differentiate between different patient groups.

The uncertainty as to the potential of naturalistic eye movement tasks as a diagnostic tool is further enhanced due to the lack of research assessing their sensitivity to differentiate between AD and preclinical decline (MCI). For naturalistic eye tracking to be an effective diagnostic tool, it must be able to differentiate those with preclinical cognitive decline from those with AD and HOC. Importantly, this review highlighted that to date, only 12 studies (30.8% of the included papers) analysed eye movement behaviours in people with MCI. Specifically, five studies analysed eye movement behaviours during a visual paired comparisons task [78,79,80,81,82], two during face processing [49,50], two during static image search [38,72], one study during reading [44], one during goal-directed paradigms with naturalistic stimuli [69], and three analysed the specificity and sensitivity of naturalistic eye movement task in the diagnosis of MCI through an AUC analysis [21,83,84].

This being said, of the limited literature analysing eye movement behaviours in people with MCI, all but one (McCade et al., [49]) observed notable differences between MCI and HOC [38,44,50,69,72,78,79,80,81,82,84]. Deficits in face memory [50], reduced novelty preference [78,79,80,81,82], increased regressions [44], and a reduced semantic interference effect [69] are all indicative of memory deficits amongst MCI populations [22]. Furthermore, impairments in scanning of both natural scenes [38] and faces [50] are indicative of an attentional deficit in selecting relevant information that occurs in MCI [22].

More significantly, only two studies [49,50] looked at MCI subgroups (aMCI and naMCI) and only one of these [49] compared the performance of these two groups on the same task. Given the increased risk of people with aMCI progressing to a diagnosis of AD, it is critical that tasks are sufficiently sensitive to differentiate between MCI subgroups, as well as between MCI, AD, and HOC more generally. Relating to this, additional studies have observed significant differences in eye movement behaviours between people with MCI and people with AD ([72,78,81] please note this paper recruited people with ‘dementia’ not AD specifically). Due to the lack of assessment of how eye movement behaviours differ in people with MCI and AD, it is somewhat difficult to draw reliable conclusions regarding the ability of eye movements to distinguish preclinical stages of cognitive impairment. However, the occurrence of significant differences in the two papers that did analyse the differences in eye movement behaviours between people AD and MCI suggests that naturalistic eye movements may be sensitive enough to differentiate AD and MCI. However, further research is required to fortify this assumption.

We have highlighted the need for further research into eye tracking during naturalistic tasks; however, specific areas show increasingly promising and robust results that are underdeveloped in the literature, which we feel require further assessment. We identified only two studies that assessed eye movements during daily living tasks, such as tea making, resulting in the research area being underdeveloped presently. Future research should strive to assess eye movements in non-lab-based settings while conducting daily living tasks which are already familiar to the participants. Further research will allow the potential of eye movements during daily living to identify cognitive impairment at clinical and preclinical stages. VPC tasks show particularly promising results for the distinction between MCI, AD, and HOC populations, and based on the papers assessed in this review, VPC tasks indicate consistent, robust, and clear markers for impairment between HOC and people with cognitive impairment. Due to this, future research should continue to assess their potential as an early indicator of cognitive impairment. Additionally, in order to truly assess the potential of eye tracking as a diagnostic tool, an AUC analysis and machine learning models should be implemented to assess the classification accuracy, sensitivity, and specificity. Therefore, we urge researchers to employ these methods when assessing naturalistic eye movements as a potential for diagnosis of cognitive impairment.

Furthermore, it is important to note that consideration of the average MMSE scores reported for the participant groups recruited indicate that people with AD were either mildly impaired or had normal cognitive functioning (see Appendix B for cognitive variables of each study). Therefore, the conclusions drawn regarding the utility of naturalistic eye movement tasks as a diagnostic tool only stands for people with mild AD. Consequently, further research recruiting those with more advanced AD is required to verify the utility of naturalistic eye movement tasks as a diagnostic and monitoring tool across all of the stages of AD.

A limitation of any review is the possibility that relevant studies may not have been captured due to limitations with the selection of databases and the search strings. However, to reduce the likelihood of missing papers, we adhered to the NIRO guidelines (V1, [37]) and consulted a librarian when producing our systematic search. Our inclusion criteria may have caused us to omit some relevant papers.

In summary, we echo the conclusions of previous reviews [36] that the potential for naturalistic eye tracking as an early diagnostic tool should not be overlooked. Over the wide range of methodologies reviewed for this paper, and the limited number of studies representing each one, noticeable patterns can be observed that suggest naturalistic eye tracking can detect changes in memory and selective attention present in the early stages of AD. Whilst traditional eye-tracking paradigms have also been demonstrated to be capable of this, the advantage of naturalistic tasks above traditional eye-tracking tasks remains that the tasks are functionally relevant and familiar. Consequently, naturalistic tasks are more ecologically valid and less stressful for older participants as they mimic activities of daily living. However, we do highlight the need for further research employing naturalistic eye-tracking tasks with a focus on their potential to distinguish MCI and preclinical stages of AD in order to allow a more accurate determination of their efficacy as an early diagnostic tool.

## Figures and Tables

**Figure 1 brainsci-11-01503-f001:**
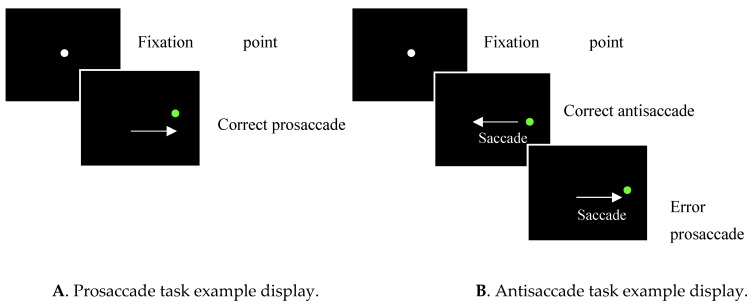
Example visual displays and eye movements patterns of a prosaccade task (Panel (**A**)), antisaccade task (Panel (**B**)), reading task (Panel (**C**)), static image search task (Panel (**D**)), visual paired comparison task (Panel (**E**)), face processing task (Panel (**F**)), and individual eye movement pattern (Panel (**G**)). Note, as studies that incorporated every-day tasks and real-life simulations did not present a fixed visual display, this task has been omitted from this figure. Panel (**F**): In each case ((**A**–**E**) above), the pattern of eye movement will consist of an individual component of saccadic eye movements with a characteristic profile, consisting of saccadic amplitude, duration, and peak velocity.

**Figure 2 brainsci-11-01503-f002:**
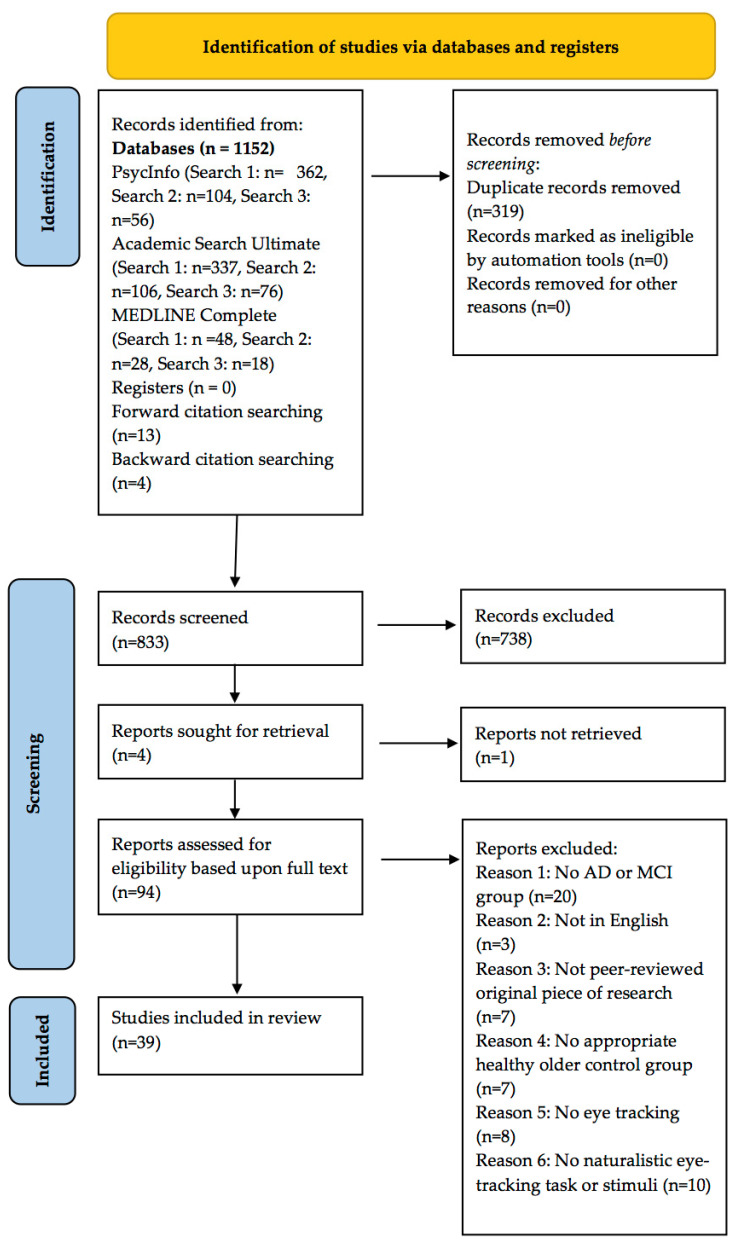
Modified PRISMA flowchart [52] detailing the number of records at each stage of the review. Note: Only the first reason for exclusion is reported for each paper (see Appendix A for full reasons for exclusion for each paper). The PRISMA flowchart [56] allows for transparent reporting of our data collection, so that our searches can be reproduced.

**Table 1 brainsci-11-01503-t001:** Quality assessment ratings using our modified version of the Downs and Black [86] checklist for each paper included in the literature review.

Reference	Quality Assessment Rating (out of 26)
Fernández et al. [39] Fernández et al. [40] Fernández et al. [41] Fernández et al. [42] Fernández et al. [43] Fraser et al. [44] Yong et al. [47] Lueck et al. [52] Mapstone et al. [46]	16 17 16 16 15 17 17 17 19
Davis and Sikorskii [64]	16
Dragan et al. [38]	19
LaBar et al. [45] Brandão et al. [63]	12 17
Mosimann et al. [65] Vallejo et al. [66] Boucart et al. [67] Boucart et al. [68]	19 18 22 19
Coco et al. [69]	20
Daffner et al. [70] Lenoble et al. [71] Oyama et al. [72] Shakespeare et al. [73] Bourgin et al. [74] Kawagoe et al. [50] McCade et al. [49] Ogrocki et al. [48] Forde et al. [25] Yong et al. [51]	20 18 16 18 20 23 19 17 13 19
Lenoble et al. [76]	19
Fernández et al. [75]	15
Crutcher et al. [79]	20
Nie et al. [82]	18
Zola et al. [81]	18
Suzuki et al. [83]	7
Chau et al. [77]	18
Haque et al. [80]	11
Lagun et al. [78]	17
Fraser et al. [84]	18
Barral et al. [85]	19

## Data Availability

As the current paper is a review, no new data were created or analysed in this study. Data sharing is not applicable to this article.

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
