# Peer review of "The Potential of Naturalistic Eye Movement Tasks in the Diagnosis of Alzheimer’s Disease: A Review"

_brainsci, 2021, doi:10.3390/brainsci11111503_

Round 1
Reviewer 1 Report
# The Potential of Naturalistic Eye Movement tasks in the Diagnosis of Alzheimer’s Disease: A Review
This is an interesting and timely review. Constrained eye movement laboratory tasks have been applied in neurodegenerative diseases over many decades. Despite that, we know little about how much of that knowledge actually generalises to the more patient-relevant domains of eye movement control in naturalistic tasks or with naturalistic stimuli.
The main methodological question in assessing this review was how comprehensive and representative a coverage of the topic it provides. To informally assess that, I went to a 2016 Current Opinion review I wrote on the general topic of eye movements in neurodegenerative conditions. Of that narrow-time-window but more general-scope review, only six papers fell into the specific domain of this manuscript. All but one were captured. The exception was this reference (although several other papers from this group were captured):
Lenoble, Q., Bubbico, G., Szaffarczyk, S., Pasquier, F., & Boucart, M. (2015). Scene categorization in Alzheimer’s disease: A saccadic choice task. Dementia and Geriatric Cognitive Disorders Extra, 5(1), 1–12. https://doi.org/10.1159/000366054
The authors have captured the classic clock face paper by Mosimannn et al. through to the more recent reading-focussed papers by the Fernández group. It seems quite thorough to me.
In the discussion section, the authors hedge that "Our inclusion criteria *may* have caused us to not include some relevant papers" and then go on to give two definite examples that had been missed. The review is not a meta-analysis, and the authors went beyond the bounds of a systematic review by manually following reference chains to identify additional papers. So I would suggest that the authors don't artificially exclude those two papers, because they did, after all identify them in some fashion. Rather, I encourage them to fold those two papers, and the additional one identified above, into the body of the narrative. The value of this paper is as a comprehensive narrative review: readers will not be concerned with the purity of whether several papers were discussed even though they hadn't been detected through the initial mechanical search strategy.
But in essence, this is a nice paper and a useful contribution to the field. At times, however, the language was difficult to parse. A number of sentences are sufficiently long and unwieldy for the subject to shift in the course of them, or for inconsistencies to develop during a sentence between singular and plural. These are minor issues and will be easy to address, however. There are a few of them to wade through, though - apologies for any pedantry here.
2.1 Data Sources
- "we did not conduct a grey literature search to increase the quality of our reviewed papers." The meaning of this quite ambiguous, with two opposite meanings: please re-phrase for clarity.
2.2.1. Inclusion Criteria
- "5 study" -> "5 studies"
Figure 2:
- The number of excluded papers seems to sum to 710, rather than the 713 given in the text?
Table 2:
- This would be more informative if the year was given for each reference, and the entries were ordered more usefully than alphabetically. e.g. Sorting by date would give an instant overview of whether these papers are becoming more common, or peaked some time ago, etc.
3.1. Reading Tasks
- In this section, the term "gaze duration" is used when I think what is meant is "fixation duration", a more accepted term and one which is used in the rest of the manuscript.
- "significantly less first-pass fixations" -> "significantly fewer first-pass fixations"
3.2. Eye Movement Behaviour during Real-life Simulations
- "contrastingly younger controls" -> "by contrast, younger controls"
- "Davis and Sikoriskii [54] observed that individuals with AD/MCI made significantly less fixations with reduced duration to salient cues compared to HOC." Here, "less" should be "fewer" but more importantly it's unclear which of two opposite meanings is intended. Did they make fewer fixations of short durations, or did they make fewer fixations, and those fixations also had short durations?
- "Comparatively, for non-salient cues, AD individuals made more fixations of comparable duration to HOC." Again, two opposite interpretations are available here. Did they make more fixations than HOC but they were of the same duration, or did they make more similar-duration fixations (compared to something unclear)?
- "individuals with AD’s eye movement search patterns were significantly more diffused than HOC."
- "Individuals with MCI’s search patterns were also more diffused than HOC"
- There are a number of sentences like this. They need to be re-phrased for readability, e.g. "the eye movement search patterns of people with AD were significantly more diffused than those of HOC"
3.3. Eye Movement Behaviours during Static Image Search
- "results in disrupted eye movements" -> "result in disrupted eye movements"
- "emotional valance" -> "emotional valence"
- "did not replicated" -> "did not replicate"
- "Thus, reducing the likelihood of error.": not a stand-alone sentence.
- "fixation entrophy" -> "fixation entropy" (this relatively uncommon term could also be defined for the reader)
- "Both individuals with MCI and HOC scan pattern similarity (between viewing and recognition phases) was higher when the scene was recognised." Re-phrase for clarity.
3.4. Eye Movement Behaviours during facial processing
- "Comparably, Ogrocki et al. [46] observed that, in general, individuals with AD fixated less on the presented faces, particularly the eyes, and spent less time looking at the different regions, and more time focusing on specific areas, of the face than HOC." Break-up/re-phrase for clarity.
3.5. Eye movement behaviours during naturalistic paradigms
- "Results indicated that individuals with AD made" -> "The person with AD made"
- "Similarly, in analysing participants eye movements" -> "Similarly, in analysing participants’ eye movements"
4. Discussion
- "the majority of papers suffered from moderate risk of bias". Th reader really needs to have it explained to them what typical sources of bias were.
- "the number of overall fixations, first-pass fixations, and skipping frequency made by individuals with AD compared to that of HOC is antithetical amongst the included studies". The meaning of this isn't clear - please expand upon this point.
-"and of these studies, and there is" -> "and among these studies, there is"
Lastly, throughout the manuscript, the term "individuals" is used when what is actually meant is simply "people". Reserve the use of the former term for when one is actually discussing within-subject effects or individual differences within groups, rather than the between-group comparisons that are largely the preserve of this paper.
Author Response
Dear Marcelina Paśko,
Please find enclosed a resubmission of the manuscript brainsci-1397737, entitled “The Potential of Naturalistic Eye Movement tasks in the Diagnosis of Alzheimer’s Disease: A Review.”, for publication in Brain sciences special edition Visuospatial Function in Early Alzheimer Disease, Healthy Elderly and MCI People.
We would like to thank you for the opportunity to resubmit this manuscript with amendments. We are grateful to you and the reviewers for the constructive comments provided to improve our manuscript. We have implemented the comments obtained and subsequently, we believe the paper has considerably improved.
Below we provide a detailed explanation of how we have addressed the comments on a point-by-point basis. The resulting amendments to the initial version of the manuscript are highlighted.
We look forward to the outcome of your review of this amended manuscript. Thank you again for your further consideration of this manuscript.
Reviewer 1
- To informally assess that, I went to a 2016 Current Opinion review I wrote on the general topic of eye movements in neurodegenerative conditions. Of that narrow-time-window but more general-scope review, only six papers fell into the specific domain of this manuscript. All but one were captured. The exception was this reference (although several other papers from this group were captured): Lenoble, Q., Bubbico, G., Szaffarczyk, S., Pasquier, F., & Boucart, M. (2015). Scene categorization in Alzheimer’s disease: A saccadic choice task. Dementia and Geriatric Cognitive Disorders Extra, 5(1), 1–12. https://doi.org/10.1159/000366054
Response: We thank you for highlighting this interesting paper that was missed in the initial version of the manuscript. The paper meets all inclusion criteria and therefore has been embedded within the manuscript.
- The authors have captured the classic clock face paper by Mosimannn et al. through to the more recent reading-focussed papers by the Fernández group. It seems quite thorough to me. In the discussion section, the authors hedge that "Our inclusion criteria *may* have caused us to not include some relevant papers" and then go on to give two definite examples that had been missed. The review is not a meta-analysis, and the authors went beyond the bounds of a systematic review by manually following reference chains to identify additional papers. So, I would suggest that the authors don't artificially exclude those two papers, because they did, after all identify them in some fashion. Rather, I encourage them to fold those two papers, and the additional one identified above, into the body of the narrative.
Response: The inclusion of these two papers in the main body of the manuscript was something the authors discussed at great lengths when writing up the manuscript. We acknowledge and agree with your reasoning for the inclusion of these papers in the main body of the manuscript, therefore we have added these two papers into the main body of the manuscript. These papers were initially omitted due to the lack of cognitive assessment. As these were the only papers to be omitted for this reason we have amended the inclusion criteria to ensure we are still robustly applying the inclusion criteria when including these papers.
“Supporting, this observation Fernández et al. [75], also observed that people with AD made more first-pass fixations compared to controls”. (Pg. 13)
“Similary, Suzuki et al [83] found that the durations of fixations across all locomotion tasks (e.g. walking through corridors, walking up or down stairs, walking through a room with or without an obstacle) did not significantly differ between the AD patient and HOC.” (Pg. 18)
- But in essence, this is a nice paper and a useful contribution to the field. At times, however, the language was difficult to parse. A number of sentences are sufficiently long and unwieldy for the subject to shift in the course of them, or for inconsistencies to develop during a sentence between singular and plural. These are minor issues and will be easy to address, however. There are a few of them to wade through, though - apologies for any pedantry here.
Response: We acknowledge that the manuscript at times contained sentences that are long and unwieldy. Therefore, we have reviewed the manuscript and amended these sentences to improve the readability of the overall manuscript. As these amendments have been made sporadically throughout the manuscript, we therefore did not include them here in the response, rather we have highlighted these amendments throughout the manuscript.
- 1 Data Sources - "we did not conduct a grey literature search to increase the quality of our reviewed papers." The meaning of this quite ambiguous, with two opposite meanings: please re-phrase for clarity.
Response: We thank you for highlighting the ambiguity of this statement, the manuscript has been amended to enhance the clarity of this statement.
“Literature that has not been published through traditional means, e.g., conference abstracts known as grey literature, is often excluded from large databases [54]. Specific grey literature searches are often conducted when collating the literature for a systematic review [55]. To the best of our knowledge, this review is the first of its kind to specifically focus on naturalistic eye movement tasks, therefore, we did not conduct a grey literature search. However, we recognise that performing grey literature searches are important to prevent publication bias and so we encourage future reviews that aim to build upon this research review to include grey literature.” (Pg. 7-8)
- 2.1. Inclusion Criteria - "5 study" -> "5 studies"
Response: We thank you for highlighting this grammatical error, due to the inclusion of additional literature in the manuscript this statement has been replaced.
“Of the 39 papers that passed through full-text screening, 27 studies had a singular AD patient group, seven studies had a singular MCI group, and five study had both an AD group and an MCI group.” (Pg. 9)
- Figure 2: - The number of excluded papers seems to sum to 710, rather than the 713 given in the text?
Response: Thank you for picking up on this discrepancy. The number of excluded papers has been corrected in the text and the table and has been updated to include the additional excluded papers from the most recent searches conducted. The number of excluded papers now stands at 738.
- Table 2: - This would be more informative if the year was given for each reference, and the entries were ordered more usefully than alphabetically. e.g. Sorting by date would give an instant overview of whether these papers are becoming more common, or peaked some time ago, etc
Response: We thank you for highlighting that the table would read more clearly with the year provided for each reference and reorganised in chronological order. We have amended the relevant table, additionally we have embedded a comment in the discussion regarding the age of the body of literature.
“Interestingly, this review highlighted that naturalistic eye movement behaviours in people with AD and MCI have gained consistent research interest from the early 2000s until the present day. Thus, highlighting the theoretical and practical relevance of the analysis of naturalistic eye movement behaviours in people with AD and MCI.”(Pg. 20)
- 1. Reading Tasks - In this section, the term "gaze duration" is used when I think what is meant is "fixation duration", a more accepted term and one which is used in the rest of the manuscript. - "significantly less first-pass fixations" -> "significantly fewer first-pass fixations"
Response: We thank you for highlighting the irregularity in the terminology used here. The manuscript has been amended to read fixation duration in all relevant circumstances. The incorrect wording in “significantly less first-pass fixations” has also been amended.
- 2. Eye Movement Behaviour during Real-life Simulations - "contrastingly younger controls" -> "by contrast, younger controls"
Response: Thank you for highlighting this grammatical error, the manuscript has been amended accordingly.
- "Davis and Sikoriskii [54] observed that individuals with AD/MCI made significantly less fixations with reduced duration to salient cues compared to HOC." Here, "less" should be "fewer" but more importantly it's unclear which of two opposite meanings is intended. Did they make fewer fixations of short durations, or did they make fewer fixations, and those fixations also had short durations?
Response: Thank you for highlighting the grammatical error and ambiguity in this sentence, the manuscript has been amended accordingly.
“Davis and Sikoriskii [64] observed that people with AD/MCI made significantly fewer fixations that were also shorter in duration to salient cues compared to HOC.” (Pg.19).
- "Comparatively, for non-salient cues, AD individuals made more fixations of comparable duration to HOC." Again, two opposite interpretations are available here. Did they make more fixations than HOC but they were of the same duration, or did they make more similar-duration fixations (compared to something unclear)?
Response: Thank you for highlighting the ambiguity in this sentence, the manuscript has been amended accordingly.
“Comparatively, for non-salient cues, people with AD made significantly more fixations than HOC. However, the durations of fixations, for non-salient cues, did not differ between AD and HOC.” (Pg. 19)
- "individuals with AD’s eye movement search patterns were significantly more diffused than HOC."- "Individuals with MCI’s search patterns were also more diffused than HOC" - There are a number of sentences like this. They need to be re-phrased for readability, e.g. "the eye movement search patterns of people with AD were significantly more diffused than those of HOC"
Response: Thank you for highlighting these grammatical errors. The phrasing of the results has been amended in the manuscript to enhance readability. Please note these overall amendments are highlighted throughout the manuscript.
- 3. Eye Movement Behaviours during Static Image Search
- "results in disrupted eye movements" -> "result in disrupted eye movements"
- "emotional valance" -> "emotional valence"
- "did not replicated" -> "did not replicate"
- "Thus, reducing the likelihood of error.": not a stand-alone sentence.
- "fixation entrophy" -> "fixation entropy" (this relatively uncommon term could also be defined for the reader)
- "Both individuals with MCI and HOC scan pattern similarity (between viewing and recognition phases) was higher when the scene was recognised." Re-phrase for clarity.
Response: We thank you for highlighting these grammatical errors and lack of clarity. The manuscript has been amended to correct these errors. Please note these overall amendments are highlighted throughout the manuscript.
- 4. Eye Movement Behaviours during facial processing
- "Comparably, Ogrocki et al. [46] observed that, in general, individuals with AD fixated less on the presented faces, particularly the eyes, and spent less time looking at the different regions, and more time focusing on specific areas, of the face than HOC." Break-up/re-phrase for clarity.
Response: Thank you for highlighting the lack of clarity in this sentence. We have rephased the sentence to improve the clarity.
“Concerning visual exploration of face stimuli, Ogrocki et al. [48] observed that, in general, people with AD fixated less on the presented faces, particularly the eye regions. People with AD also spent less time exploring different facial regions, and rather spent more time focusing on specific areas of the face than HOC.”(Pg. 18)
- 5. Eye movement behaviours during naturalistic paradigms - "Results indicated that individuals with AD made" -> "The person with AD made" - "Similarly, in analysing participants eye movements" -> "Similarly, in analysing participants’ eye movements"
Response: We thank you for highlighting these grammatical errors. The manuscript has been amended to correct these errors. Please note these overall amendments are highlighted throughout the manuscript.
- "the majority of papers suffered from moderate risk of bias". The reader really needs to have it explained to them what typical sources of bias were.
Response: We thank you for highlighting the uncertainty as to the sources of bias. The manuscript has been amended to clarify what bias refers to here and we have provided some examples of sources of bias.
“Here, bias refers to factors that can systematically affect the observations and conclusions of a study [94]. Subsequently, some examples of sources of bias include problems with the comparability of the criteria used to select samples [selection bias; 95], problems with the measurement of outcomes [detection bias; 94] and problems with whether research is published or not [publication bias; 96]. Interestingly, the three most common sources of bias found within the included papers were: failure to describe the characteristics of participants lost to exclusion, failure to take into account participants lost to exclusion in analyses, and no justification for sample sizes (see supplementary materials for quality assessment ratings of each paper). Therefore, as many of the papers included here suffer from moderate to high risk of bias, a certain level of caution should be assumed when considering the potential application of the findings highlighted here” (Pg.21)
- "the number of overall fixations, first-pass fixations, and skipping frequency made by individuals with AD compared to that of HOC is antithetical amongst the included studies". The meaning of this isn't clear please expand upon this point.
Response: We apologise for the unclear inclusion of the term ‘antithetical’, here we were stating that the results obtained regarding the number of overall fixations, first-pass fixations, and skipping frequency made by people with AD compared to that of HOC, are inconsistent. Therefore, this phrasing has been amended in the manuscript to improve clarity.
“Specifically, concerning reading paradigms, the number of overall fixations, first-pass fixations, and skipping frequency made by people with AD compared to that of HOC is inconsistent amongst the included studies” (Pg. 22)
- "and of these studies, and there is" -> "and among these studies, there is"
Response: We thank you for highlighting this grammatical error, the manuscript has been amended to correct this.
“Further, this review highlighted just three studies which utilised facial stimuli and among these studies, and there is a high degree of variability in their findings.” (Pg. 22)
- Lastly, throughout the manuscript, the term "individuals" is used when what is actually meant is simply "people". Reserve the use of the former term for when one is actually discussing within subject effects or individual differences within groups, rather than the between-group comparisons that are largely the preserve of this paper.
Response: Thank you for highlighting the incorrect use of the term individuals the manuscript has been amended to replace the incorrect word with “people”. Please note these overall amendments are highlighted throughout the manuscript.
Reviewer 2
- When reading the title ‘The potential of naturalistic eye movement tasks in the diagnosis of Alzheimer’s disease’ I expected to find some studies in this review, which actually attempted to differentiate between Alzheimer’s patients and controls on an individual level, reporting sensitivity and specificity of identifying someone as a patient. As far as I understood, all studies included in this review are basic research, reporting differences in the group means or medians. This limitation of the reviewed studies should be added to the discussion.
Response: We thank you for acknowledging that the title of this review is somewhat misleading. Relating to your suggestion, the initial literature search did indeed highlight several papers that employed area under the curve (AUC) analyses to assess the specificity and sensitivity of naturalistic eye tracking tasks in the differentiation of people with AD, MCI and healthy controls. However, these papers were omitted from the original manuscript as they did not report patterns of eye movement behaviours. Thus, to address the somewhat misleading nature of the title we have revisited the initial screening log and subsequently embedded papers focusing on AUC analyses in the main body of the manuscript. Here, we have presented the results in a specifically dedicated diagnostic utility section of the results “3.5 Analyses of the specificity and sensitivity of eye-movements in diagnostic practices”.
“3.5 Analyses of the specificity and sensitivity of eye-movements in diagnostic practices
Previous research has demonstrated the potential of machine learning to aid in the screening and early diagnosis of neurodegenerative disorders [92] Subsequently, machine learning models built on naturalistic eye tracking data from people with AD and MCI could offer a non-invasive screening tool to aid with early detection of cognitive impairment. In this current work we identified five papers [85, 84, 78, 80, 81] that utilised machine learning techniques and conducted an AUC analysis to decipher the specificity and sensitivity of naturalistic eye movement tasks in differentiating people with AD, MCI and HOC.
Considering the utility of reading tasks, Fraser et al [84] tracked participants eye movements whilst reading, either silently or aloud, before they completed a comprehension task. Fraser et al [84] found that the best classification result, achieved by combining (eye tracking, speech, comprehension questions measures; AUC = 0.88, accuracy = 0.83) outperforms a classifier trained on neuropsychological tests (AUC=0.75, accuracy-0.65). Thus, indicating that eye tracking and audio recording during reading tasks could aid in the classification of cognitive impairment and may prove more successful than current neuropsychological tests.
Alternatively, considering static image search tasks Barral et al [85] asked people with AD and HOC to perform the Cookie Theft Picture Description Task. This required participants to scan a line drawing and verbally describe the scene while their eye movements and speech were recorded. Interestingly, here Barral et al. [85] observed that eye tracking data combined with machine learning models can successfully distinguish people with AD and HOC (AUC=.73). This model was further improved by combining the eye tracking and speech data (AUC=.80).
Lagun et al [78] assessed people with AD, MCI and HOC on a VPC task using abstract images. From this Lagun et al. [78] found that when fixations, saccades, and re-fixations during the VPC task are modelled in tandem with the Support Vector Machines (SVMs) algorithm, people with MCI can be distinguished from HOC with 87%, sensitivity of 97% and specificity of 77%. Consequently, this study provides strong support that eye-movement patterns during VPC tasks can distinguish people with MCI and HOC and that machine learning could aid in the automatic detection of cognitive impairment.
Utilising a longitudinal methodology, Zola et al. [81] analysed whether VPC is reflective of cognitive decline. Specifically, AUC analysis showed that all but one participant who had a novelty preference of less than 50% on the task at initial testing changed in their diagnosis within the 3 year interval of testing. Participants who scored between 50% and 67% were at less risk. Critically, those who scored more than 67 were at a zero risk of further cognitive decline regardless of whether they were initially categorised as HOC or aMCI. Therefore, the VPC task had the capability to predict participants who would change in their diagnosis (regardless of whether they were HOC or aMCI) before diagnosis was changed clinically. Critically, when assesing novelty preference after either a 2 second or 2 minutie delay, Nie et al [82] AUC analysis showed that novelty preference scores of 0.605 in the 2 minute delay task could effectively distinguish MCI and HOC (70% accuracy, 72% specificity, and 53% sensitivity). In a 12-month follow-up, 9 participants had progressed to MCI. Those participants whose novelty preference score fell below the 0.605 cut-off point at initial testing showed significantly greater cognitive decline at the 12 month follow-up.
Similar to the VPC task, Haque et al [80] assessed people with AD, MCI and HOC on a visual- spatial memory task in which a familiarised presented image was altered by the removal or addition of an item. Using MoCA scores as a comparison, Haque et al. [80] found that performance on the visual-spatial memory task achieved an AUC of 0.85 (sensitivity = 0.83, specificity =0.74). Moreover, when compared with disease status the model achieved an AUC of 0.85, sensitivity of 0.85, and specificity of 0.75. Overall, the above studies appear to provide support that performance on naturalistic eye tracking tasks can be aid in the classification and identification of AD and MCI status with high sensitivity.” (Pg. 19-20)
- In the Discussion the authors pointed out that studies, which assessed two specific cognitive functions, namely memory recognition and selective attention were most ‘successful’ at distinguishing between AD and controls. The authors also mention that the specific task the participants need to perform in an experiment influences the result. With regards to future research, which tasks and parameters have the best chances to lead to a distinction between patients and controls? This could be added to a ‘future research’ section.
Response: Thank you for this suggestion. We have added a short “future research” section to the discussion. Here we discuss more specific avenues for future research and tasks that show particularly promising and robust markers for the distinction between patients and controls. We also discuss the areas within the literature which could most benefit from future research due to the current underdevelopment of that area.
“We have highlighted the need for further research into eye tracking during naturalistic tasks however specific areas show increasingly promising and robust results that are underdeveloped in the literature, which we feel require further assessment. We identified only two studies that assessed eye movements during daily living tasks such as tea making, resulting in the research area being currently underdeveloped. Future research should strive to assess eye movements in non-lab-based settings while conducting daily living tasks, which are already familiar to the participants. Further research will allow the potential of eye movements during daily living to identify cognitive impairment at clinical and pre-clinical stages. VPC tasks show particularly promising results for the distinction between MCI, AD, and HOC populations and based on the papers assessed in this review, indicate consistent, robust and clear markers for impairment between HOC and people with cognitive impairment. Due to this future research should continue to assess their potential as an early indicator of cognitive impairment. Additionally, in order to truly assess the potential of eye tracking as a diagnosis tool, AUC analysis and machine learning models should be implemented to assess classification accuracy, sensitivity and specificity. Therefore, we urge researchers to employ these methods when assessing naturalistic eye movements as a potential for diagnosis of cognitive impairment.” (Pg. 23-24)
- In line 48 the authors write ‘As attention and saccades are thought to recruit overlapping brain regions, …’. As the term ‘saccades’ is limited to high velocity eye movements only, it does not seem to be appropriate here. I suggest to use the term ‘oculomotor control’ or something more general.
Response: We thank you for highlighting that the word ‘saccade’ is inappropriate in this context, we have amended the manuscript to include as per your suggestion oculomotor control.
“As attention and oculomotor control..”. (Pg.2)
- In the second part of the sentence the authors write: ‘saccades are likely to be disturbed by cognitive impairment that occurs in neurodegenerative disorders’. It could be helpful to shortly mention what kind of impairment occurs before this sentence.
Response: As you have highlighted the second part of this sentence is somewhat ambiguous. Subsequently, the manuscript has been amended to improve the clarity of the point raised here.
“Current evidence suggests that attention is the first non-memory domain to be affected in AD [4]. As attention and oculomotor control are thought to recruit overlapping brain regions [12], saccades are likely to be disturbed by the reductions in inhibitory control and executive function that occur in neurodegenerative disorders [13]. (Pg. 2)
- In line 98 ‘Wilcoxon et al. has demonstrated’. The word ‘has´ should be deleted.
Response: This additional irrelevant word has been removed from the manuscript.
“Wilcockson et al. [35] demonstrated that the anti-saccade task can distinguish between MCI subgroups. People with AD and aMCI showed slower latencies and higher error rates than people with naMCI and HOC, and people with aMCI performed more similarly to people with AD than people with naMCI or HOC” (Pg.6)
- Materials and Methods section: The paper is missing a short explanation of the NIRO guidelines, the PRISMA guidelines and the Downs and Black’s tool. These should be added.
Response: On reflection the omission of short explanation of the NIRO and PRISMA guidelines, and the Downs and Black’s tool is a limitation of the manuscript in its current form. Therefore, the manuscript has been amended to incorporate short descripts of the above.
“The NIRO systematic review guidelines [V1; 37] comprise a comprehensive checklist to follow when conducting and writing a review of non-interventional research to ensure transparency and reduce bias.” (Pg. 7)
“The PRISMA flowchart [56] allows for transparent reporting of our data collection, so that our searches can be reproduced.” (Pg. 11)
“The Downs and Black checklist [86] is designed to allow the quality assessment of interventional research, including assessments of internal and external validity, reporting, and power.” (Pg. 11)
- In line 325 you mention that in the Brandao et al. study ‘individuals with AD fixate on the screen more when visual cues are present’. I did not understand where else they could look. Can you please explain the task of this study shortly?
Response: We acknowledge that this sentence as it stands is highly ambiguous. Therefore, the sentence has been amended to enhance clarity.
“In general, people with AD fixated their gaze on the screen (as opposed to looking at the experimenters' face) more when visual cues were present, irrespective of their relevance.” (Pg. 15)
- In line 329 and 330 you mention ‘diffused’ eye movements. Can you define the term ‘diffused’ in this context?
Response: We thank you for highlighting the ambiguity of the word ‘diffused’ in this context. Here diffused is referring to scan patterns being less focused, therefore, we have amended the manuscript to include less focused in place of ‘diffused’.
e.g. “During free image viewing, the fixation patterns of HOC were significantly less focused than people with MCI (higher fixation entropy), thereby suggesting that HOC had a wider spread of attention than people with MCI.” (Pg. 15).
Please note. The above is one example of this alteration, as the overall amendments made to address this point are dispersed throughout the manuscript, we have highlighted these alterations.
- It is not completely clear to me what the results were in the Forde et al. study. The sentence ‘Results indicated that individuals with AD made comparable number of fixations of fixation duration to YC and HOC’ (line461-462) does not have a proper sentence structure, but is crucial to understand the study.
Response: The terminology used to introduce the results in this section is misleading and has led to confusion. As a result, we have altered the terminology used to introduce the results obtained by Forde et al.
“Interestingly, Forde et al. [25] observed that the person with AD made comparable number of fixations of equivalent fixation duration to younger and older controls. More specifically, the proportion of task-relevant and task-irrelevant fixations did not differ between the HOC, young control and people with AD. In HOC, young controls and people with AD 10-15% of fixations were to relevant objects that were to be used in the next stage of the tea-making tasks.” (Pg. 18)
- At times, eye movement deficits in AD and MCI are related to the cognitive deficits. It seemed to me that a short summary of the most important cognitive deficits could be included in the introduction, preferably early on before the review of the saccade and antisaccade task.
Response: On reflection, the inclusion of a short summary of the most important cognitive deficits improves the readability of the overall manuscript. Therefore, the manuscript has been amended to include a short description of the cognitive deficits occurring in AD.
“For example, overall deficits in memory (e.g. recalling recent events), and more specific deficits in language, semantic memory, attention, and visuospatial function characteristically occur in AD [4, 5, 6, 7].” (Pg. 1)
- Differences in the difficulty of the tasks are mentioned in the passing. However, they might be more prominent in the introduction and in the discussion. In the introduction, it seems to me that an important advantage of artificial laboratory tasks pose the additional problem that (some) AD and MCI patients face problems in the implementation and execution of a new task. Reading, tea making and other well practiced tasks do not need to be newly instructed, and thus assessment of eye movements in these practiced tasks is way easier, and possibly also more informative. I leave it to the authors whether they like to include this motivation for using natural and naturalistic tasks in the assessment of AD and MCI. However, in the discussion, this should be a topic, and it should be discussed whether natural tasks might differ in difficulty, and whether this relates to the results.
Response: Thank you for this suggestion, we have added more detail regarding this in the introduction and have discussed this in more detail in the discussion section.
“In addition, novel lab-based tasks require participants to quickly adapt, follow instructions and learn new behaviours to complete these tasks successfully. Critically, there are many factors, such as age, sex, intelligence and motivation, that may influence an individual's ability to learn new behavior [26]. Intuitively, these factors are likely to influence both neurotypical people and people with neurological impairment, particularly in early stages of the task. Subsequently, altered eye movement behaviours may reflect a lack of task understanding rather than the presence or absence of a cognitive disorder. In contrast, it is likely that naturalistic tasks such as reading or tea making will already be familiar tasks to participants, and therefore require little to no explanation of how to complete the task. This removes the increased level of difficulty of having to learn a new task and reduces the likelihood of misunderstanding the task instructions.” (Pg. 6 )
Discussion
“However, some of these inconsistencies may be explained by methodological variations, for example the active opposed to passive nature of the tasks applied in Davis and Sikoriskii and Mapstone et al. [64, 46]. Similarly, concerning static image search methodologies, the only study that required the participant to passively view stimuli with no additional goal-directed task was also the only study to observe no significant differences in the eye movement behaviour between people with AD and HOC [45]. This further highlights and supports the reasoning that the differences in results may be due to varying methodologies. Therefore, eye movement variations may be an artefact of the task employed as opposed to the insensitivity of naturalistic eye movement tasks. The current literature review included multiple studies with varying methodologies that differ in their complexity and task difficulty. Goal-directed; unfamiliar tasks are likely to prove more taxing than free-viewing tasks particularly for individuals with cognitive impairment. Further, unfamiliar, lab-based assessments require the participant to first understand the task instructions and then quickly learn how to perform the task successfully, increasing the difficulty and complexity of the task. Familiar everyday tasks such as reading, tea-making, and free-viewing of scenes do not require this learning process and allow for a more natural assessment of participants eye movements. However, results from reading and tea-making tasks may not be sufficiently sensitive to distinguish between people with AD and HOC [25]. The increased level of complexity of goal-directed eye movement tasks may be required to robustly identify cognitive impairment in pre-clinical stages” (Pg. 22 )
- On line 502: "central targets"
Response: Thank you for highlighting this grammatical issue, the manuscript has been amended to correct this.
“alongside a decreased ability to detect targets during visual search [63, 38, 66, 67, 68, 65, 70, 71, 72, 73]” (Pg. 21)
Response:
- In the end of the discussion, the authors mention a previous review by Seligman & Giovanetti (2015). I think the review should be mentioned and shortly summarized in the introduction. In the discussion I think it is adequate to compare results and conclusions from the present review with the previous review. It would also worth mentioning why the authors decided to do an own review (put otherwise: what are the differences between the present and the previous review), and how much the reviewed studies overlapped.
Response: We thank you for highlighting that greater consideration of the Seligman & Giovanetti (2015) review would be beneficial to the flow of the introduction. On revisiting this review several interesting aspects were picked up on that have fortified the arguments posed for the utility of naturalistic eye tracking tasks. The manuscript has been amended to reflect your suggestion.
“Considering the promise of naturalistic eye movement tasks in the diagnosis of disorders of ageing, a somewhat recent review has concluded that naturalistic eye movement tasks have the potential to successfully differentiate healthy older adults from people with MCI [36]. Specifically, Seligman and Giovanetti [36] highlighted that important eye movement patterns, including fixation location, duration and saccade magnitude, are highly consistent in HOC, and therefore, are sensitive enough to highlight meaningful alterations indicative of MCI. However, the review focused primarily on MCI studies and excluded a number of domains, including the literature on reading..” (Pg. 6)
“Moreover, although considering the same topic area, Seligman and Giovanetti [36] focused on the theoretical utility of naturalistic eye movement tasks in people with MCI; therefore, the overlap with the present review is minimal” (Pg. 6-7).
Reviewer 3
- The naturalistic eye movement tasks were defined as “tasks undertaken in a normal daily life setting” (ll. 190-193). Although at the first sight such definition might seem appealing, in the current paper it resulted in inclusion of the tasks with quite particular eye movement patterns, such as reading, and, on the other hand, it excluded the naturalistic viewing tasks that are widely used for AD diagnostics, such as the Visual Paired Comparison (VPC) task.
Specifically, the eye movement pattern in reading (8 included papers) involves quick systematic switches between left-to-right and right-to-left eye movements as well as gaze transitions between regularly spaced locations defined by the text lines. Such pattern is extremely distinctive and it is hardly compatible with the eye movement patterns in visual search or face perception or natural behavior tasks, included in the current review. On the contrary, both familiarization and test phases of the Visual Paired Comparison task involve free viewing exploration, with the eye movement patterns similar to that in visual search. Visual Paired Comparison is a common task used for AD diagnostics, e.g.: Crutcher et al. 2009; Lagun et al. 2011; Zola et al. 2012; Whitehead et al. 2018; Nie et al. 2020. Surprisingly, many of these studies were detected in the initial selection, as indicated in the supplementary Excel file, but were later rejected because “No naturalistic task”.
Response: We thank you for highlighting that the definition of naturalistic eye movement tasks prescribed to in this manuscript has resulted in the inconsistent inclusion/omission of relevant literature, specifically literature relating to visual comparison tasks. We appreciate the reasoning provided for the inclusion of Visual paired comparison tasks and therefore as per your suggestion, we have revisited the initial screening log and amended the manuscript to include those papers were previously omitted. Moreover, we conducted an additional literature search incorporating the search strings “VPC” OR “paired comparison*” OR “paired-comparison*” OR “free view*” OR “free-view*” OR “visual scan*” OR ((“natural” OR “scene”) AND (“view*” OR “vision”)) to ensure that all papers relating to visual paired comparison tasks were captured. Please note. Here we have not included Whitehead et al., 2018 as they did not present visual stimuli for a minimal duration of 5s or include naturalistic stimuli.
“A final search was conducted with search terms associated with visual paired-comparison and free-viewing tasks. This final search was conducted on 19th October 2021.” (Pg. 7)
“The visual paired comparisons (VPC) task has a proven sensitivity to memory decline [89]. Typically during the VPC task participants are first presented with a visual stimulus for a fixed period of time (familiarization phase). Following a delay, participants are presented with a pair of stimuli, one that is the same as the familiarization stimulus and one that is new [test phase; 90]. As participants are not instructed where to direct their gaze during both the familiarization and test phases, participants will engage in free visual search. Consequently, even if the visual stimuli presented are artifical (e.g. line drawings) the visual search strategy engaged by the participant is naturalistic by nature.
Six of the studies included employed VPC comparison tasks. More specifically, of these studies two included naturalistic visual stimuli, two included artificial stimuli, and two analysed VPC performance longitudinally using articifcal stimuli. Both Chau et al [77] and Lagun et al [78] assessed performance on the VPC task, incoporating artificial stimuli. Chau et al [77] first presented participants with a slide containing four novel images. This was followed by two further slides containing two novel images and two repeated images. Relative fixation time was calculated by dividing the fixation time to the novel images by the total fixation time for all four slide images. In doing so, Chau et al [77] found that people with AD showed lower relative fixation times when viewing novel images than on repeated images compared to HOC. In addition reduced relative fixation time was associated with lower MMSE task scores. Interestingly, Lagun et al [78] also assessed VPC task performance in people with MCI. From this Lagun et al [78] found that VPC performance can effectively distinguish between people with AD, MCI, and HOC. Specifically, machine learning demonstrated an accuracy of 87%, sensitivity of 97% and specificity of 77% when distinguishing participant groups.
In contrast both Crutcher et al [79] and Haque et al [80] incorporated images of naturalistic scenes during the VPC. Assessing people with MCI and HOC, Crutcher et al [79] varied the delay interval (2 seconds or 2 minutes) between the initial viewing of the image and the test trial (in which the repeated image and novel image were presented simultaneously). Interestingly, at the 2 second delay participants’ viewing behaviour was comparable; a novel image preference of 71% was observed across groups. However, when the delay between images increased to 2 minutes, the viewing preference for the novel image was significantly reduced only in people with MCI. This finding demonstrates that a delay period during the VPC highlights viewing pattern distinctions in cognitively impaired populations compared to healthy adults.
Haque et al [80] looked at VPC in people with AD, MCI, and HOC. Participants were asked to view coloured images of naturalistic scenes with no explicit instructions. After initial viewing, participants were presented with the image once again but with either an item removed from the scene or an item added to the scene. The ROI were defined as the location the item was removed or added to. For people with AD and MCI, the time spent viewing the ROI and number of fixations to the ROI was significantly lower compared to HOC. Thus, there were clear performance differences between cognitively impaired individuals and HOC. Results from the above studies indicate that visual scanning behaviour, specifically novelty preference, varies between HOC and people with AD and MCI, highlighting key and robust markers for cognitive impairment.
Two studies utilising a VPC methodology analysed performance longitudinally [81, 82]. In doing so, both Zola et al. [81] and Nie et al. [82] corroborate with the aforementioned findings that fixation duration on novel stimuli was significantly shorter in MCI and AD than HOC. Echoing the findings of Crutcher et al. [79], Nie et al. [82] found novelty preference only differed significantly between people with MCI and HOC when the delay period was 2 minutes, but not 2 seconds. Moreover, Nie et al. [82] found that this difference remained significant at a two week follow-up.” (Pg. 16-17)
- The inclusion criteria lacks essential restrictions, what resulted in inclusion of the studies where eye movement patterns were far from naturalistic. Particularly, there was no limit on the minimal duration of visual exploration. For example, in the included studies by Boucart, Calais et al. (2014), Lenoble et al. (2018), and Bourgin et al. (2018) the stimuli were displayed just for 1 s. I doubt that such short presentation allowed eye movements that may be called naturalistic.
Response: We indeed agree that the minimal duration for visual exploration is a particularly important characteristic to consider when analysing naturalistic eye movement tasks. As a result, we have amended the definition of naturalistic tasks prescribed to here to state that a minimal duration of 5s visual exploration was required for the task to be classified as naturalistic by nature. However, in the interest of completeness rather than removing studies that presented stimuli for a shorter duration from the manuscript, we have made a clear distinction between studies that employ tasks that are naturalistic by nature (e.g. free visual search with a minimal duration of visual exploration of 5s) and studies that recruit tasks that are artificial (e.g. stimuli presentation for 1s) with naturalistic stimuli (e.g. images of natural scenes).
“Here naturalistic eye movement tasks were defined as those tasks that either (a) incorporate goal-directed paradigms with naturalistic stimuli (e.g. a prosaccade task in which participants are instructed to perform a saccade towards an object within a naturalistic scenes), (b) tasks in which stimuli was presented for a minimal duration, 5s, that enabled participants to engage in free unrestricted visual exploration (e.g. unrestricted static image search and visual paired comparison tasks) or (c) tasks that are the same as (e.g. making a cup of tea or navigating an environment) or closely mirrored (e.g. virtual reality) tasks undertaken in a normal daily life setting.” (Pg. 8)
- In several selected studies participants got explicit instructions for their eye movements. For example, in both studies by Boucart et al. (2014) “participants were asked to saccade to the picture containing an animal”. Bourgin et al. (2018) used the prosaccade and antisaccade tasks, which can be hardly considered as naturalistic. This is discussed in Introduction (ll. 105-110) and, more than that, the critique of these tasks as lacking ecological validity is a starting point for introduction of the alternative, naturalistic tasks, but nevertheless the study by Bourgin et al. (2018) is included.
Deliberating limitations of the review the authors admitted: “the task itself remained lab-based and contrived” (l. 514). For the purpose of AD diagnostics, the tasks most likely should be lab-based. But it is unclear why the studies with pre-instructed eye movements were not excluded if the inclusion criterion was “tasks undertaken in a normal daily life setting” (ll. 190-193).
Response: Whilst initially it may appear that the studies incorporating pre-instructed eye movement tasks are ‘artificial tasks’ the physical execution of these task is reflective of naturalistic eye movements individuals perform daily. That is, individuals naturally perform reflexive saccades towards objects or are asked to “look at this”, or “look over here”, thereby performing a prompted saccade, often in their day-to-day lives. Therefore, the authors here argue that prosaccade tasks incorporating naturalistic stimuli facilitate the performance of an eye-movement that is characteristically naturalistic. Conversely, the eye movements required by the anti-saccade task is highly artificial, regardless of whether the stimuli are naturalistic or not the anti-saccade task is still artificial. Subsequently, the authors elected to include pre-instructed prosaccade tasks but exclude anti-saccade tasks. Considering the manuscript in its initial format, this resulted in the partial exclusion of results obtained by Bourgin et al (2018). Specifically, results that relating to the antisaccade task were removed whilst results pertaining to the prosaccade task remain in the manuscript. Moreover, an explanation as to why the authors deemed these tasks to be naturalistic has been embedded in the manuscript.
“In this review we deemed prosaccade tasks as a naturalistic paradigm as they replicate eye movements frequently performed in daily life. For example, if individuals are asked to “look at this” or “look over here” they subsequently perform a prompted goal-directed saccade similar to that employed in prosaccade tasks. In contrast, antisaccade tasks were excluded from this review due to antisaccade eye movements being artificial by nature and unintuitive..” (Pg. 8-9)
- By amazing coincidence, just a couple of weeks ago I reviewed the review paper on the very similar topic for another journal. It was an expert review from the well-known group and I, as well as other reviewers, evaluated it highly. Naturally, during reviewing the current paper I have compared it with that review and this comparison is not in favour of the current paper. In my view, the main problem of the current paper is a mechanistic application of the protocol of systematic reviews. This protocol is attractive because it maximizes unbiased selection, but it may be suboptimal for the translational, non-interventional research topics because it does not allow sufficient flexibility of selection and cannot replace expert knowledge of the field.
Response: The quality of literature reviews relies upon comprehensive, systematic, and transparent identification of all the relevant literature (Topor et al., 2021, Ioannidis, 2016; Moher et al., 2009; Siddaway et al., 2019). Systematic reviews follow a procedure that attempts to identify, appraise and synthesise all empirical evidence that meets a specific inclusion criterion to answer a highly focused, resultantly, systematic reviews are widely considered as the ‘gold-standard’ format of review (Impellizzeri, 2012; Munn, 2018; Dissemination, C. F. R. A., 2009). Subsequently, several guidelines exist to help researchers achieve these standards, and some of the guidelines are now endorsed or mandated by some journals and collaborative groups specialised in conducting systematic and literature reviews, such as the EQUATOR network (UK EQUATOR Centre, Centre for Reviews and Disseminations (University of York), Campbell Collaboration (Campbell Collaboration), and Cochrane Collaboration (The Cochrane Collaboration).
Although, reviews of non-interventional research aim to provide an explanatory framework of empirical phenomena (Glass, 1972), the quality of the review still relies upon the comprehensive, systematic, and transparent identification of all the relevant literature (Topor et al., 2021). Subsequently, we elected to follow the recently developed NIRO guidelines (Topor et al., 2021) to ensure the quality of the review.
We do, however, acknowledge that the strict adherence to guidelines cannot and will never replace expert knowledge of the field. As a result, this current review was guided and overseen by Professor T. Crawford, who himself has worked with eye-tracking paradigms in AD populations for more than two decades. Moreover, on occasion we did deviate from the strict systematic review guidelines and show flexibility in that we conducted additional searches incorporating additional search strings following team discussions in which it was acknowledged that we may have missed relevant literature. This, flexibility has again been shown in that as per your suggestion we have conducted another supplementary search incorporating specific strings to ensure we captured relevant literature regarding Visual Paired Comparison tasks.
- In my view, the definition of naturalistic eye movement tasks should include free viewing behavior with a certain minimal duration (e.g., 5 s), when participants have sufficient time for unrestricted visual exploration. After initial selection of the studies, any constraint on eye movement behavior imposed by image features or by task demands should be carefully checked in each study. I believe that such approach could result in a large, homogenous and informative collection of studies, which allows to reach more refined conclusions than that in the current paper.
Response: We thank you for highlighting this limitation and acknowledge that the definition of naturalistic eye movement tasks prescribed to in the initial version of the manuscript resulted in the inconsistent inclusion/omission of relevant literature.
When reviewing the results of the literature search it appeared that studies typically either (a) employed paradigms that are predominantly laboratory constructs (goal-directed saccades) with naturalistic stimuli (e.g. tasks in which participants are told how to direct their gaze, for example, to perform a prosaccade towards an object in a naturalistic scene) or (b) employed tasks which are naturalistic by nature (e.g. free visual search). Whilst initially it may appear that the former (a) tasks are still ‘manufactured or artificial tasks’ the physical execution of this task is something that individuals perform daily. That is, it is very common in everyday life for individuals to naturally perform reflexive saccades towards objects or be asked to “look at this”, or “look over here”, thereby performing a prompted saccade. Therefore, the authors here argue that prosaccade tasks incorporating naturalistic stimuli are still in some sense “naturalistic”. Conversely, the eye movement required by the anti-saccade task originally emerged from Hallet’s lab as powerful tool to probe prepotent inhibitory control, but with no pretence towards ecological validity. Therefore, in our view irrespective of the nature of the stimulus, the origins emerged as specific laboratory based tool rather a form of naturalistic behaviour. To ensure this review is as comprehensive as possible, we included research that employed naturalistic eye movements (e.g. free visual search) and goal directed eye movements with naturalistic stimuli (e.g. prosaccade tasks with naturalistic stimuli). Therefore, we have amended the definition of naturalistic tasks, and the subsequent literature search, within manuscript.
However, we acknowledge that this alteration of the definition results in literature that falls within four somewhat distinct domains; studies incorporating reading tasks, studies incorporating goal directed eye movements with naturalistic stimuli, studies incorporating naturalistic tasks, and studies analysing eye movements during day to day activities. As a result of the distinct nature of these domains of research drawing parallels between the obtained results is neither valid nor useful. Resultantly, we have highlighted this within the manuscript and presented the results in line with these somewhat distinct domains.
“Here naturalistic eye movement tasks were defined as those tasks that either (a) incorporate goal-directed paradigms with naturalistic stimuli (e.g. a prosaccade task in which participants are instructed to perform a saccade towards an object within a naturalistic scenes), (b) tasks in which stimuli was presented for a minimal duration, 5s, that enabled participants to engage in free unrestricted visual exploration (e.g. unrestricted static image search and visual paired comparison tasks) or (c) tasks that are the same as (e.g. making a cup of tea or navigating an environment) or closely mirrored (e.g. virtual reality) tasks undertaken in a normal daily life setting..” (Pg. 8).
“This literature review revealed that studies examining naturalistic eye movements in people with AD, MCI, and healthy older controls can be broadly classified into four domains; reading tasks, goal-directed paradigms with naturalistic stimuli (e.g. goal-directed saccades towards naturalistic stimuli), paradigms that are naturalistic by nature (e.g. free image viewing or visual paired comparisons), and paradigms including or simulating everyday activities (e.g. making a cup of tea or navigating an environment). Importantly, the eye movement behaviours facilitated by these four domains of literature are somewhat distinct. That is, during reading tasks participants typically perform highly specialised eye movement patterns including saccades, fixations and regressions mediated by the text they are reading [87]. These highly specialised eye movement patterns are largely distinct from the free exploratory saccades and fixations typically performed during free visual search tasks. Due to the distinct nature of these domains of research, drawing parallels between the obtained results is somewhat difficult and arguably invalid. Subsequently, the results of such studies will be presented separately..” (Pg. 12)
Reviewer comment 6
The description of the prosaccade and antisaccade tasks occupies a disproportionately large space in Introduction and is even supported by a figure. As I mentioned above, these tasks are far from naturalistic, i.e., opposite to what the review is focused on. So I’d suggest to reduce the description. Instead, it might be helpful to add a figure with examples of eye movement patterns in the included naturalistic tasks in order to emphasize their similarity.
Response: We thank you for highlighting that the description of the pro- and anti-saccade tasks occupied a disproportionally large space in the introduction. First, we have amended this description to be more succinct. In addition we have amended the embedded figure to highlight the similarity of the eye movements engaged in naturalistic tasks and the pro- and anti-saccade tasks.
“The prosaccade task, requires participants to perform rapid, reactive saccades towards a suddenly appearing target from a central fixation point [14, See panel A figure 1]. Interestingly, some evidence has shown that the latency of saccades produced by people with AD are longer than healthy older controls [HOC; 15]. However, alternative research has found no differences in saccadic latency between people with AD and HOC [See 15 for review]. Due to these inconsistencies, it appears that prosaccade tasks alone are not sufficiently sensitive to function as an AD diagnostic tool.
Conversely, the anti-saccade task has yielded more consistent results. This task requires participants to inhibit a reactive saccade towards a target and instead perform a saccade towards the opposite target absent location [16; See panel B figure 1]. Specifically, whilst anti-saccade latencies do not appear to differentiate between people with AD and HOC [15], the frequency of inhibition errors made on the anti-saccade task is significantly higher in those with AD [11, 17, 18]. Moreover, the frequency of inhibition errors on the anti-saccade task has been found to be predictive of dementia severity [11, 14, 17]. Furthermore, whilst HOC correct a large proportion of anti-saccade task errors, people with AD often fail to do this, resulting in a higher number of uncorrected errors than HOC [10, 11, 17, 19, 20]. The homogeneity demonstrated in the literature suggests that the anti-saccade task may be a valid AD diagnostic tool [21].” (Pg. 2)
Reviewer comment 7
- 5: the inclusion criterion #3 is “3) the study included an AD/MCI group without comorbidities or other neurological disorders”. It is not clear how comorbidities or other neurological disorders (except related to the oculomotor control system) can preclude naturalistic eye movement behaviour.
Response: Previous research has shown that comorbidities are somewhat common in AD. For example, many individuals with AD present with comorbid anxiety disorders (Ferretti et al., 2001), and around 30% of individuals with AD display comorbid depression (Santiago & Potashkin, 2021). Moreover, some individuals with alternative neurological conditions, including Parkinson’s disease (PD; Emre, 2003) and Multiple Sclerosis (MS; Rogers, & Panegyres, 2007), present with comorbid cognitive impairment and in some cases comorbid dementia. Interestingly, additional research has shown that these comorbidities and alternative neurological disorders can independently substantially influence naturalistic eye movement behaviours. For example, a somewhat recent meta-analysis concluded that individuals with depression show reduced maintenance of gaze towards positive stimuli, and anxious individuals showed difficulty disengaging from threatening stimuli during visual search tasks (Armstrong & Olatunji, 2013). Moreover, Waldthaler et al., (2018) observed that individuals with PD fixation on words for a greater duration and make a greater number of regressions when reading. As these morbidities and alternative neurological disorders can independently influence naturalistic eye movement behaviours, if one were to analyse naturalistic eye movement behaviours in individuals with AD with these comorbidities, it would be difficult to parse apart the influence of AD from the influence of the comorbidity. As this review sought to analyse the potential utility of naturalistic eye movement tasks in the specific diagnosis of AD/ MCI it was important to ensure that the papers analysed did not recruit participants with these comorbidities.
We acknowledge that this reasoning had not been highlighted in the original manuscript, therefore, the manuscript has been amended to highlight the reasoning for this exclusion criteria.
“Previous research has shown that many individuals with AD present with comorbidities. For example, many individuals with AD have anxiety disorders [57], and around 30% of individuals with AD present with comorbid depression [58]. Moreover, some individuals with alternative neurological conditions, including Parkinson’s disease [PD; 59] and Multiple Sclerosis [60], present with comorbid cognitive impairment and in some cases comorbid dementia. Interestingly, additional research has shown that these comorbidities and alternative neurological disorders can independently substantially influence naturalistic eye movement behaviours. For example, a somewhat recent meta-analysis concluded that individuals with depression show reduced maintenance of gaze towards positive stimuli, and anxious individuals showed difficulty disengaging from threatening stimuli during visual search tasks [61]. Moreover, Stock et al. [62] observed that individuals with PD fixate on words for a greater duration and make a greater number of regressions when reading. As these morbidities and alternative neurological disorders can independently influence naturalistic eye movement behaviours, if one were to analyse naturalistic eye movement behaviours in individuals with AD with these comorbidities, it would be difficult to parse apart the influence of AD from the influence of the comorbidity. As this review sought to analyse the potential utility of naturalistic eye movement tasks in the specific diagnosis of AD/ MCI, it is important to strive to reduce the likelihood of including participants with these comorbidities.” (Pg. 7-8)
Reviewer comment 8
- 462: “number of fixations of fixation duration” misprint?
Response: We thank you for highlighting that a word has been omitted from this sentence, the manuscript has been amended to correct this omission.
“Interestingly, Forde et al. [25] observed that the person with AD made comparable number of fixations of equivalent fixation duration to younger and older controls.” (Pg. 17-18)
574: “all bar one” but?
Response: We thank you for highlighting this grammatical error, the manuscript has been amended to correct this.
“This being said, of the limited literature analysing eye movement behaviours in people with MCI, all but one [McCade et al., 49] observed notable differences between MCI and HOC [79, 78, 81, 80, 82, 50, 38, 72, 44, 69,84 ].”(Pg. 22)
We thank you again for the opportunity to revise our manuscript. Please do let us know if you have any additional questions or comments. We look forward to hearing from you in due course.
Kind Regards,
Megan Rose Readman
ESRC NWSSDTP PhD Candidate
C-8, Fylde College
Department of Psychology
Lancaster University
Lancaster LA1 4YF

Reviewer 2 Report
Reviewer comments
The potential of naturalistic eye movement tasks in the diagnosis of Alzheimer’s disease
This manuscript presents a review of the research literature concerning the potential of using eye movements as a method to diagnose Alzheimer’s disease and mild cognitive impairment (MCI) under naturalistic viewing conditions. After a summary of the relevant results from classical laboratory tasks, in particular saccade and antisaccade tasks, the authors review the results of 22 of studies, most of which are reading tasks, in addition to static image search and face viewing, plus two studies that report real-life simulations. They draw the conclusion that naturalistic eye tracking could measure subtle changes in cognitive functioning.
The excellently organized review is very well written, and metriculously conducted and documented. It presents an excellent overview of the topic of naturalistic eye movements, and a balanced discussion of the results. The review has the prospect of being very useful for clinical researchers and eye-movement researchers interested in clinical applications. However, I have some suggestions for improvements.
- When reading the title ‘The potential of naturalistic eye movement tasks in the diagnosis of Alzheimer’s disease’ I expected to find some studies in this review, which actually attempted to differentiate between Alzheimer’s patients and controls on an individual level, reporting sensitivity and specificity of identifying someone as a patient. As far as I understood, all studies included in this review are basic research, reporting differences in the group means or medians. This limitation of the reviewed studies should be added to the discussion.
- In the Discussion the authors pointed out that studies, which assessed two specific cognitive functions, namely memory recognition and selective attention were most ‘successful’ at distinguishing between AD and controls. The authors also mention that the specific task the participants need to perform in an experiment influences the result. With regards to future research, which tasks and parameters have the best chances to lead to a distinction between patients and controls? This could be added to a ‘future research’ section.
- In line 48 the authors write ‘As attention and saccades are thought to recruit overlapping brain regions, …’. As the term ‘saccades’ is limited to high velocity eye movements only, it does not seem to be appropriate here. I suggest to use the term ‘oculomotor control’ or something more general.
- In the second part of the sentence the authors write: ‘saccades are likely to be disturbed by cognitive impairment that occurs in neurodegenerative disorders’. It could be helpful to shortly mention what kind of impairment occurs before this sentence.
- In line 98 ‘Wilcoxon et al. has demonstrated’. The word ‘has´ should be deleted.
- Materials and Methods section: The paper is missing a short explanation of the NIRO guidelines, the PRISMA guidelines and the Downs and Black’s tool. These should be added.
- In line 325 you mention that in the Brandao et al. study ‘individuals with AD fixate on the screen more when visual cues are present’. I did not understand where else they could look. Can you please explain the task of this study shortly?
- In line 329 and 330 you mention ‘diffused’ eye movements. Can you define the term ‘diffused’ in this context?
- It is not completely clear to me what the results were in the Forde et al. study. The sentence ‘Results indicated that individuals with AD made comparable number of fixations of fixation duration to YC and HOC’ (line461-462) does not have a proper sentence structure, but is crucial to understand the study.
- At times, eye movement deficits in AD and MCI are related to the cognitive deficits. It seemed to me that a short summary of the most important cognitive deficits could be included in the introduction, preferably early on before the review of the saccade and antisaccade task.
- Differences in the difficulty of the tasks are mentioned in the passing. However, they might be more prominent in the introduction and in the discussion. In the introduction, it seems to me that an important advantage of artificial laboratory tasks pose the additional problem that (some) AD and MCI patients face problems in the implementation and execution of a new task. Reading, tea making and other well practiced tasks do not need to be newly instructed, and thus assessment of eye movements in these practiced tasks is way easier, and possibly also more informative. I leave it to the authors whether they like to include this motivation for using natural and naturalistic tasks in the assessment of AD and MCI. However, in the discussion, this should be a topic, and it should be discussed whether natural tasks might differ in difficulty, and whether this relates to the results.
- On line 502: "central targets"
- In the end of the discussion, the authors mention a previous review by Seligman & Giovanetti (2015). I think the review should be mentioned and shortly summarized in the introduction. In the discussion I think it is adequate to compare results and conclusions from the present review with the previous review. It would also worth mentioning why the authors decided to do an own review (put otherwise: what are the differences between the present and the previous review), and how much the reviewed studies overlapped.
Author Response

(The authors gave the same response as above.)

Reviewer 3 Report
The paper presents the systematic review of application of eye tracking in naturalistic viewing conditions to diagnostics of Alzheimer’s Disease (AD). The authors strictly follow the systematic review guidelines in searching, screening and selection of studies. Six inclusion criteria were set and 28 papers that satisfied these criteria were selected. The results of these papers were summarized and discussed. The authors pointed out some potential of naturalistic eye tracking as a diagnostic tool for AD but also expressed their reservations related to large variations in the results among the selected studies.
I appreciate the main goal of this review to explore the utility of ecologically valid, naturalistic viewing tasks for diagnostics of AD. But the inadequate definition of the inclusion criteria (the section 2.2.1), particularly the definition of a naturalistic viewing task, resulted in selection of the inhomogeneous, inconsistent collection of the studies. This led to the vague conclusions that rather hide than elucidate the potential of the naturalistic eye tracking for AD diagnostics.
Major points
- The naturalistic eye movement tasks were defined as “tasks undertaken in a normal daily life setting” (ll. 190-193). Although at the first sight such definition might seem appealing, in the current paper it resulted in inclusion of the tasks with quite particular eye movement patterns, such as reading, and, on the other hand, it excluded the naturalistic viewing tasks that are widely used for AD diagnostics, such as the Visual Paired Comparison (VPC) task.
Specifically, the eye movement pattern in reading (8 included papers) involves quick systematic switches between left-to-right and right-to-left eye movements as well as gaze transitions between regularly spaced locations defined by the text lines. Such pattern is extremely distinctive and it is hardly compatible with the eye movement patterns in visual search or face perception or natural behavior tasks, included in the current review. On the contrary, both familiarization and test phases of the Visual Paired Comparison task involve free viewing exploration, with the eye movement patterns similar to that in visual search. Visual Paired Comparison is a common task used for AD diagnostics, e.g.: Crutcher et al. 2009; Lagun et al. 2011; Zola et al. 2012; Whitehead et al. 2018; Nie et al. 2020. Surprisingly, many of these studies were detected in the initial selection, as indicated in the supplementary Excel file, but were later rejected because “No naturalistic task”.
- The inclusion criteria lacks essential restrictions, what resulted in inclusion of the studies where eye movement patterns were far from naturalistic. Particularly, there was no limit on the minimal duration of visual exploration. For example, in the included studies by Boucart, Calais et al. (2014), Lenoble et al. (2018), and Bourgin et al. (2018) the stimuli were displayed just for 1 s. I doubt that such short presentation allowed eye movements that may be called naturalistic.
- In several selected studies participants got explicit instructions for their eye movements. For example, in both studies by Boucart et al. (2014) “participants were asked to saccade to the picture containing an animal”. Bourgin et al. (2018) used the prosaccade and antisaccade tasks, which can be hardly considered as naturalistic. This is discussed in Introduction (ll. 105-110) and, more than that, the critique of these tasks as lacking ecological validity is a starting point for introduction of the alternative, naturalistic tasks, but nevertheless the study by Bourgin et al. (2018) is included.
Deliberating limitations of the review the authors admitted: “the task itself remained lab-based and contrived” (l. 514). For the purpose of AD diagnostics the tasks most likely should be lab-based. But it is unclear why the studies with pre-instructed eye movements were not excluded, if the inclusion criterion was “tasks undertaken in a normal daily life setting” (ll. 190-193).
By amazing coincidence, just a couple of weeks ago I reviewed the review paper on the very similar topic for another journal. It was an expert review from the well-known group and I, as well as other reviewers, evaluated it highly. Naturally, during reviewing the current paper I have compared it with that review and this comparison is not in favor of the current paper. In my view, the main problem of the current paper is a mechanistic application of the protocol of systematic reviews. This protocol is attractive because it maximizes unbiased selection, but it may be suboptimal for the translational, non-interventional research topics because it does not allow sufficient flexibility of selection and cannot replace expert knowledge of the field.
In my view, the definition of naturalistic eye movement tasks should include free viewing behavior with a certain minimal duration (e.g., 5 s), when participants have sufficient time for unrestricted visual exploration. After initial selection of the studies, any constraint on eye movement behavior imposed by image features or by task demands should be carefully checked in each study. I believe that such approach could result in a large, homogenous and informative collection of studies, which allows to reach more refined conclusions than that in the current paper.
Minor points
The description of the prosaccade and antisaccade tasks occupies a disproportionately large space in Introduction and is even supported by a figure. As I mentioned above, these tasks are far from naturalistic, i.e., opposite to what the review is focused on. So I’d suggest to reduce the description. Instead, it might be helpful to add a figure with examples of eye movement patterns in the included naturalistic tasks in order to emphasize their similarity.
p. 5: the inclusion criterion #3 is “3) the study included an AD/MCI group without comorbidities or other neurological disorders”. It is not clear how comorbidities or other neurological disorders (except related to the oculomotor control system) can preclude naturalistic eye movement behavior.
l. 462: “number of fixations of fixation duration” misprint?
l. 574: “all bar one” but?
Author Response

(The authors gave the same response as above.)

Round 2
Reviewer 3 Report
I appreciate the great work of the authors to address my concerns. I have no objections against publishing the paper.
I hope that the displacement of the panels of figures 1 and 2 will be corrected in the final version.